# JAK2 inhibition mediates clonal selection of RAS pathway mutations in myeloproliferative neoplasms

Nabih Maslah [1,2,8], Nina Kaci[3,8], Blandine Roux[3,8], Gabriela Alexe [4], Raphael Marie[3], Hélène Pasquer[3,5], Emmanuelle Verger [1,2], Rafael Daltro De Oliveira[5], Cécile Culeux[3], Bochra Mlayah[2], Nicolas Gauthier[5], Fanny Gonzales[4], Lin-Pierre Zhao [6], Saravanan Ganesan [2], Panhong Gou[2], Frank Ling[3], Juliette Soret-Dulphy[5], Nathalie Parquet[6], William Vainchenker [6], Emmanuel Raffoux [6], Rose Ann Padua[2], Stéphane Giraudier[1,2], Caroline Marty [7], Isabelle Plo[7], Camille Lobry [3], Kimberly Stegmaier [4], Alexandre Puissant [3], Jean-Jacques Kiladjian[2,5,9], Bruno Cassinat [1,2,9] & Lina Benajiba [3,5,9] ✉

JAK (Janus Kinase) inhibitors, such as ruxolitinib, were introduced a decade ago for treatment of myeloproliferative neoplasms (MPN). To evaluate ruxolitinib's impact on MPN clonal evolution, we interrogate a myelofibrosis patient cohort with longitudinal molecular evaluation and discover that ruxolitinib is associated with clonal outgrowth of RAS pathway mutations. Single-cell DNA sequencing combined with ex vivo treatment of *RAS* mutated CD34⁺ primary patient cells, demonstrates that ruxolitinib induces *RAS* clonal selection both in a JAK/STAT wild-type and hyper-activated context. *RAS* mutations are associated with decreased transformation-free and overall survival only in patients treated with ruxolitinib. In vitro and in vivo competition assays demonstrate increased cellular fitness of *RAS*-mutated cells under ruxolitinib or *JAK2* knock-down, consistent with an on-target effect. MAPK pathway activation is associated with *JAK2* downregulation resulting in enhanced oncogenic potential of *RAS* mutations. Our results prompt screening for pre-existing *RAS* mutations in JAK inhibitor treated patients with MPN.

Targeted therapies have emerged over the last two decades as a fundamental component of the anti-cancer therapeutic arsenal. Their use is often associated with resistance mechanisms mainly driven by cancer cell adaptability[1]. In the case of chronic myeloid leukemia, for example, despite the efficacy of BCR-ABL1 tyrosine kinase inhibitors, resistance mechanisms driven by the selection of on-target *ABL1* mutated clones appear in a subset of patients[2]. Given the intra-patient heterogeneity of cancer, clonal selection of pre-existing resistant subclones without second-site mutations in the targeted protein also emerged as a potential resistance mechanism. Indeed, *FLT3* secondary

[1]Université Paris Cité, APHP, Hôpital Saint-Louis, Laboratoire de Biologie Cellulaire, Paris, France. [2]INSERM UMR 1131, Institut de Recherche Saint-Louis, Paris, France. [3]INSERM UMR 944, Institut de Recherche Saint-Louis, Paris, France. [4]Department of Pediatric Oncology, Dana-Farber Cancer Institute and Boston Children's Hospital, Boston, MA, USA. [5]Université Paris Cité, APHP, Hôpital Saint-Louis, Centre d'Investigations Cliniques, INSERM, CIC 1427 Paris, France. [6]Université Paris Cité, APHP, Hôpital Saint-Louis, Département d'hématologie et d'Immunologie, Paris, France. [7]INSERM UMR 1287, Gustave Roussy, Université Paris-Saclay, Villejuif, France. [8]These authors contributed equally: Nabih Maslah, Nina Kaci, Blandine Roux. [9]These authors jointly supervised this work: Jean-Jacques Kiladjian, Bruno Cassinat, Lina Benajiba. ✉e-mail: lina.benajiba@inserm.fr

mutations are infrequently found in patients developing secondary resistance to the FLT3 inhibitors gilteritinib and crenolanib. Acquired resistance is mediated by subclonal selection of RAS/MAPK activating mutations[3,4]. Similarly, *MEK1* and *NRAS* mutated clonal selection underlies resistance to the RAF inhibitor vemurafenib in melanoma[5,6]. Modulation of clonal architecture thus appears as a major resistance mechanism, creating new oncogenic dependencies and therapeutic opportunities across many cancer types. A better understanding of treatment-induced clonal evolution is needed to improve cancer patients' outcomes.

JAK-targeted therapies, such as the first-in-class JAK1/2 ATP competitive inhibitor ruxolitinib[7–9], were introduced a decade ago for the treatment of myeloproliferative neoplasms (MPN) driven by JAK/STAT signaling activating mutations in *JAK2*[10], *CALR*[11] or *MPL*[12]. Additional mutations in genes involved in the regulation of epigenetic (*ASXL1*, *EZH2*, *TET2*, *DNMT3A*, *IDH1/2*), mRNA splicing (*SRSF2*, *SF3B1*, *U2AF1*, *ZRSR2*), gene transcription (*TP53*, *NFE2*, *IKZF1*) or non-JAK/STAT signaling pathways (*NRAS*, *KRAS*, *CBL*)[13,14] complement the MPN molecular landscape. Several studies highlighted the role of such mutations in modifying patient prognosis[15,16], leading to their implementation in advanced prognostic scoring systems[17]. MPNs are considered preleukemic, and clonal evolution with sequential acquisition of mutations is likely to participate in the transition towards a more aggressive disease[13,18,19]. In addition to high molecular risk (HMR) mutations (*ASXL1*, *EZH2*, *IDH1/2*, *SRSF2*, *U2AF1*), reported to be associated with adverse outcomes, mutations in *TP53* are also frequently associated with the occurrence of secondary AML[13,20]. No clear mechanism explaining the selection of mutated clones has been identified to date.

Despite significant quality of life improvement with decreased symptoms and reduced splenomegaly, the effects of ruxolitinib on disease modification and long-term patient outcome remain controversial[21]. Although *JAK2* mutations within the kinase domain have been described as driving ruxolitinib resistance in vitro, we and others failed to identify second-site on-target mutations involving *JAK2* in patients[22–24]. MPNs inter-patient molecular landscape is highly heterogeneous, with some patients harboring a unique clone, while others carrying a number of genetically diverse sub-clones[15,18]. Longitudinally, the emergence of a new clone or an increase in the allele frequency of pre-existing clones, sometimes at the expense of other sub-clones, is observed[13]. We thus hypothesized that modulation of clonal architecture upon ruxolitinib selective pressure could not only hamper the clinical efficacy of JAK inhibitors, but also adversely modulate the natural history of MPNs.

In this study, to evaluate the effect of JAK inhibitors on MPN clonal evolution and its potential impact on clinical outcome, we interrogate a cohort of patients with myelofibrosis and validate our findings using in vitro and in vivo MPN models. We reveal a selection of *RAS* mutations upon ruxolitinib exposure, negatively impacting clinical outcome. Mechanistically, this effect results from the enhanced oncogenic potential of *RAS*-mutated clones after inactivation of the JAK/STAT pathway. Using MPN as a dynamically evolving model, our results reflect the major challenge of selectively targeting heterogeneous and adaptive diseases such as cancer.

## Results
### Ruxolitinib treatment is associated with the accumulation of RAS pathway mutations in patients with myelofibrosis
To evaluate the impact of ruxolitinib treatment on clonal evolution, we collected clinical and molecular data across a cohort of 143 patients with the myelofibrosis MPN subtype. Within this monocentric cohort, 72 patients were treated with ruxolitinib. Patients' characteristics are presented in Table S1. Ruxolitinib-treated patients had more constitutional symptoms, higher IPSS and DIPSS scores, and a higher proportion of HMR mutations (Table S1). Non-ruxolitinib-treated patients were diagnosed between 1990 and 2019, while ruxolitinib-

treated patients were diagnosed between 1994 and 2018. 73% ($n = 52/71$) and 47% ($n = 34/72$) patients were diagnosed after ruxolitinib EMA approval for myelofibrosis treatment (in 2012 or later), in the non-ruxolitinib and ruxolitinib-treated groups, respectively. Treatment of patients not exposed to ruxolitinib is detailed in Table S2. Molecular data obtained using a Next Generation Sequencing (NGS) panel targeting 36 genes involved in myeloid diseases, detailed in the methods section, is presented in Supplementary Data 1.

First, we focused on the 73 myelofibrosis patients for whom longitudinal molecular evaluation was available. We analyzed molecular evolution on all patients exposed ($n = 45$) or not exposed ($n = 28$) to ruxolitinib between baseline and follow-up molecular evaluation (Fig. 1A). After a median molecular follow-up of 21 months IQR[13; 44] in the ruxolitinib exposed population, a total of 53 mutations were newly detected among 26 patients. 28% ($n = 15/53$) of these mutations occurred within genes involved in the RAS pathway (*NRAS*, *KRAS*, *CBL*) (Fig. 1B). Additionally, if we only focus on the 22 patients for whom the baseline NGS was performed prior to ruxolitinib initiation ($n = 22/45$), 37% of the newly detected mutations involved RAS pathway genes after a median ruxolitinib exposure time of 16 months IQR[12; 28] (Fig. S1A). In the non-ruxolitinib exposed population, 12 patients accumulated a total of 14 mutations over a molecular follow-up of 25 months IQR[15; 55]. Only one of these mutations (7%) occurred in RAS pathway genes (Fig. 1C). Using a COX regression analysis, ruxolitinib exposure was associated with *RAS* mutations accumulation (HR 9.8 CI$_{95\%}$[1.23; 78.89], $p = 0.031$) (Fig. 1D **and** Table S3). Conversely, age and disease aggressiveness markers including an intermediate-2 or high DIPSS prognosis score at time of molecular evaluation and presence of HMR (*ASXL1*, *EZH2*, *SRSF2*, *U2AF1* or *IDH1/2*) mutations, were not associated with RAS pathway mutations accumulation (Table S3).

We then compared *RAS* mutations variant allele frequency between baseline and follow-up molecular evaluations according to patients' exposures to ruxolitinib. *RAS* mutations variant allele frequency significantly increased at follow-up molecular evaluation only in patients exposed to ruxolitinib (Fig. 1E, F). Accordingly, at last molecular evaluation, RAS pathway mutations were more frequently present in patients treated with ruxolitinib, including patients harboring multiple *RAS*-mutated clones (Fig. 1G, H). Importantly, while 10 *ASXL1* mutations ($n = 10/53$, 19%) were newly detected among ruxolitinib exposed patients in comparison with only 1 acquired *ASXL1* mutation ($n = 1/14$, 7%) in the non-ruxolitinib exposed population (Fig. 1B, C); *ASXL1* mutations variant allele frequency did not increase at follow-up molecular evaluation (Fig. S1B, C). This observation suggests a specific effect of ruxolitinib on *RAS* pathway mutations clonal selection.

### RAS pathway mutations adversely impact MPN patient prognosis in the context of ruxolitinib treatment
Because clonal selection of additional mutated clones is usually associated with worse prognosis in myelofibrosis, and RAS pathway mutations have been associated with poor prognosis and poor ruxolitinib treatment response in MPN[25,26], we next aimed to evaluate the clinical impact of ruxolitinib-induced *RAS* mutant clonal selection. We interrogated our cohort of 143 patients with myelofibrosis according to their exposure to ruxolitinib (Fig. 2A). Within a median follow-up of 7.0 years, 13 and 7 patients experienced a transformation of their MPN to MDS/AML, and 26 and 9 patients died, among ruxolitinib treated ($n = 72$) and not treated ($n = 71$) patients respectively.

To identify variables associated with Overall survival (OS) and Transformation Free Survival (TFS), we performed a Cox model univariate analysis including age, DIPSS at molecular evaluation, presence of RAS pathway mutations, but also potential confounding molecular variables such as the presence of HMR (*ASXL1*, *EZH2*, *SRSF2*, *U2AF1* or *IDH1/2*) mutations. In the overall cohort, presence of *RAS* mutations was associated with decreased OS and TFS in univariate analysis (HR

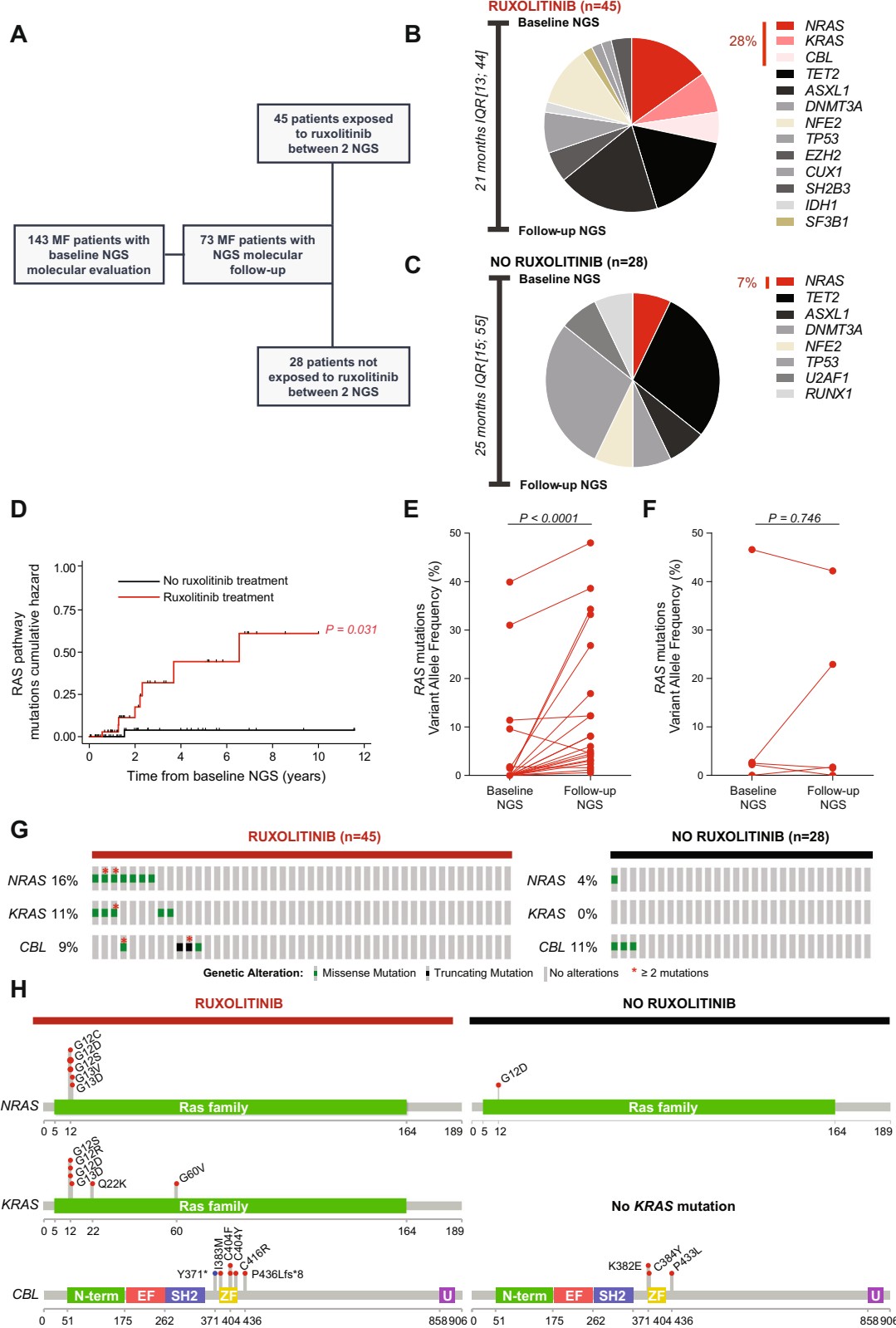

3.2 CI$_{95\%}$[1.6; 6.4], $p = 0.001$ and 5.2 CI$_{95\%}$[2.1; 12.5], $p < 0.001$). *RAS* mutational status remained significantly associated with TFS but not with OS in the multivariate analysis (3.0 CI$_{95\%}$[1.2; 7.8], $p = 0.023$ and $p = 0.0884$) (Fig. S1D and Tables S4-5). Whereas the presence of *RAS* mutations was independently associated with decreased transformation free (HR 6.6 CI$_{95\%}$[1.9; 23.1], $p = 0.003$) and overall (HR 3.1 CI$_{95\%}$[1.3; 7.1], $p = 0.008$) survival in ruxolitinib treated patients (Fig. 2B

and Tables S6, 7), such mutations did not influence clinical outcome of MPN patients not treated with ruxolitinib (Fig. 2C). In the ruxolitinib treated cohort, median OS and TFS were respectively 18.51 years and not reached in the absence of RAS pathway mutations, and 7.63 years and 7.54 years when a RAS pathway mutation was present. Mean ruxolitinib treatment duration until transformation was 44.7 months $_{Range}$[4.2; 81.5] in the global cohort, 33.5 months $_{Range}$[4.2; 81.5]

**Fig. 1 | Ruxolitinib treatment is associated with the accumulation of RAS signaling pathway mutations in patients with myelofibrosis. A** Flow-chart depicting the patients included in the study according to their exposure to ruxolitinib treatment and molecular data availability. **B, C** Pie Charts depicting the additional mutations longitudinally newly identified in patients with myelofibrosis exposed ($n = 45$) (**B**) or not exposed ($n = 28$) (**C**) to ruxolitinib between baseline and follow-up molecular evaluation. **D** Cumulative Hazard curve of the acquisition of RAS pathway mutations for ruxolitinib-treated patients ($n = 45$) compared to non-ruxolitinib-treated patients ($n = 28$) with molecular follow-up available. Two-sided COX proportional hazards regression was used for comparing the groups. P-value reported in the figure. **E, F** RAS mutations variant allele frequency longitudinal

evolution in patients with myelofibrosis exposed ($n = 45$) (**E**) or not exposed ($n = 28$) (**F**) to ruxolitinib between baseline and follow-up molecular evaluation. Variant allele frequency was considered zero when mutations were absent at baseline or follow-up molecular evaluation. Statistical significance determined using two-sided Mann-Whitney test in comparison to baseline molecular evaluation. P-values reported in the figure. **G, H** Oncoprints (**G**) and lollipop plots (**H**) showing RAS pathway mutations in patients treated with ruxolitinib ($n = 45$) compared to patients not treated with ruxolitinib ($n = 28$). *N-term = N-terminal domain 1, EF = EF hand-like domain, SH2 = SH2-like domain, ZF=Zinc finger, C3HC4 type (RING finger), U = UBA/TS-N domain.* Source data are provided as a Source Data file.

for patients harboring a *RAS* mutation, and 62.5 months $_{Range}$[36.7; 77.3] for patients without *RAS* mutation. Given the intrinsic biases associated with comparing ruxolitinib-treated and non-treated patients, due to the clinical use of ruxolitinib in patients with more aggressive disease, we also evaluated the impact of *RAS* mutational status on OS and TFS after stratification according to the DIPSS score. Importantly, the presence of *RAS* mutations remained associated with a poorer OS and TFS among patients harboring an intermediate-2/high risk score when exposed to ruxolitinib (HR 3.8 CI$_{95\%}$[1.3; 11.0], $p = 0.012$ and HR 6.2 CI$_{95\%}$[1.6; 24.2], $p = 0.009$) (Fig. S1E). Conversely, these associations were not significant among intermediate-2/high risk score patients not exposed to ruxolitinib (HR 1.6 CI$_{95\%}$[0.2; 14.3], $p = 0.682$ for OS and HR 2.0 CI$_{95\%}$[0.2; 19.5], $p = 0.559$ for TFS). A low number of *RAS* mutations and death/transformation events precluded a similar analysis among DIPSS low/intermediate-1 patients.

Among ruxolitinib treated patients harboring RAS pathway mutations, 47.06% ($n = 8/17$) transformed to AML/MDS, while only 16.67% ($n = 1/6$) of *RAS* mutated patients not treated with ruxolitinib developed an acute transformation (Fig. 2D). Variant allele frequency of the *RAS* mutated clones increased at time of transformation for all ruxolitinib treated patients with available molecular evaluation ($n = 5$ patients), while it did not increase in the *RAS* mutated patient who transformed in the absence of ruxolitinib treatment (Fig. 2E). The molecular landscape at time of transformation was available for 8 patients and is presented in Fig. S1F according to ruxolitinib treatment status. While 7 of these 8 patients harbored an *ASXL1* mutation at the time of transformation, including the patient who transformed in the absence of ruxolitinib exposure, the variant allele frequency of *ASXL1* mutations only increased in 1 patient (Fig. 2F). This finding argues again for the specific effect of ruxolitinib on RAS pathway mutations selection.

Causes of death in non-ruxolitinib-treated patients included hemorrhage, infection, and global condition deterioration, with 4 patients who died in an AML/MDS disease stage, while 5 died in chronic MPN (Table S8).

Taken together, our results suggest that ruxolitinib-induced selection of RAS pathway mutations represents an unfavorable prognostic factor associated with poor clinical outcome, including both a decreased OS and increased MDS/AML transformation risk. We also unmask a differential prognosis associated with *RAS* mutations according to ruxolitinib treatment status, suggesting that ruxolitinib exposure might increase *RAS*-mutated clones oncogenic fitness.

### Ruxolitinib treatment positively selects *RAS* mutant clones both in the context of JAK/STAT hyper-activation and with wild-type (WT) JAK/STAT

To evaluate the mechanistic link between ruxolitinib exposure and RAS pathway mutations selection, we next investigated whether ruxolitinib positively selects *RAS* mutant clones through ex vivo treatment of primary human CD34$^+$ hematopoietic cells sorted from peripheral blood mononuclear cells of patients with *RAS*-mutated MPN ($n = 6$). *RAS* mutation allele frequency increased across all six patients upon

in vitro ruxolitinib treatment (Fig. 3A). In contrast, driver mutations' (*JAK2, CALR,* or *MPL*) allele frequencies decreased upon ruxolitinib exposure in five out of six patients, suggesting that *RAS*-mutated clones might not always be present within the MPN driver clone (Fig. 3A). Evolution of the variant allele frequencies of the remaining mutations are presented in Fig. S2. Of note, although no mutation was found in the RAS pathway regulator *PTPN11* across our cohort, ruxolitinib ex vivo treatment of our primary patient samples revealed a *PTPN11$^{Q510E}$* low VAF mutation that was not detected prior to ruxolitinib (Patient#2, Fig. S2). To study the clonal architecture of these samples, we performed single-cell DNA sequencing using a custom-made targeted panel. *RAS* mutations were either present within the MPN driver clone, or in a clone without a driver mutation, suggesting that ruxolitinib-induced *RAS* clonal selection can occur both in a JAK/STAT hyper-activated or WT context (Fig. 3B). Furthermore, longitudinal molecular evaluation of two of these patients treated with ruxolitinib for their MPN showed an increase in the allele frequency of *RAS* mutations over time, confirming in vivo in patients the impact of ruxolitinib treatment on *RAS* mutations selection. Notably, in one of these patients chronic MPN transformed into a secondary myelodysplastic syndrome harboring a dominant *KRAS* mutated clone (Fig. 3C).

### MAPK pathway activation confers a fitness advantage and increased clonogenic potential to *JAK2$^{V617F}$*, *CALR$^{del52}$* and WT hematopoietic cells upon ruxolitinib exposure

To further validate the effects of ruxolitinib on *Jak2$^{WT}$* hematopoietic cells, we next performed competitive co-culture experiments using murine lineage negative (Lin$^-$) bone marrow cells isolated from CD45.1/2 *Jak2$^{WT}$ Nras$^{G12D}$* transgenic mice or from syngeneic CD45.1 *Jak2$^{WT}$ Nras$^{WT}$* C57BL/6 mice. Starting from a 50/50 or a 20/80 *Nras* mutant to WT cellular ratio, we observed a time-dependent enrichment in mutant cells at the expense of WT cells, suggesting that *Nras$^{G12D}$* cells are less sensitive than normal cells to the JAK inhibitor's anti-proliferative effect (Fig. 4A, S3A). Indeed, when untreated, *Nras$^{WT}$* cells had a higher proliferation rate than *Nras$^{G12D}$* cells. However, while *Nras$^{WT}$* cells were sensitive to ruxolitinib treatment, *Nras$^{G12D}$* cells displayed an increased proliferation rate after ruxolitinib exposure in comparison with non-treated *Nras$^{G12D}$* cells (Fig. 4B). Consistently, an in vivo bone marrow transplantation competition experiment further validated the increased fitness advantage of CD45.1/2 *Jak2$^{WT}$ Nras$^{G12D}$* cells in comparison with CD45.2 *Jak2$^{V617F}$ Nras$^{WT}$* cells upon ruxolitinib selective pressure. A mixture of 10% *Jak2$^{WT}$ Nras$^{G12D}$* lineage negative cells and 90% *Jak2$^{V617F}$ Nras$^{WT}$* cells were co-transplanted within lethally irradiated secondary C57BL/6 recipient mice, to mimic the clonal selection situation represented by Patient#1 in Fig. 3C, in which the *RAS* mutation was present in a non-driver subclone in competition with the main *RAS$^{WT}$JAK2$^{V617F}$* driver clone. Ruxolitinib treatment was initiated 4 weeks after bone marrow transplantation (Fig. 4C). CD45.1/2 and CD45.2 peripheral blood flow cytometry monitoring showed an increased proportion of *Nras$^{G12D}$* mutant cells at the expense of *Nras$^{WT}$* cells in the ruxolitinib-treated group in comparison with vehicle-treated mice (Fig. 4D).

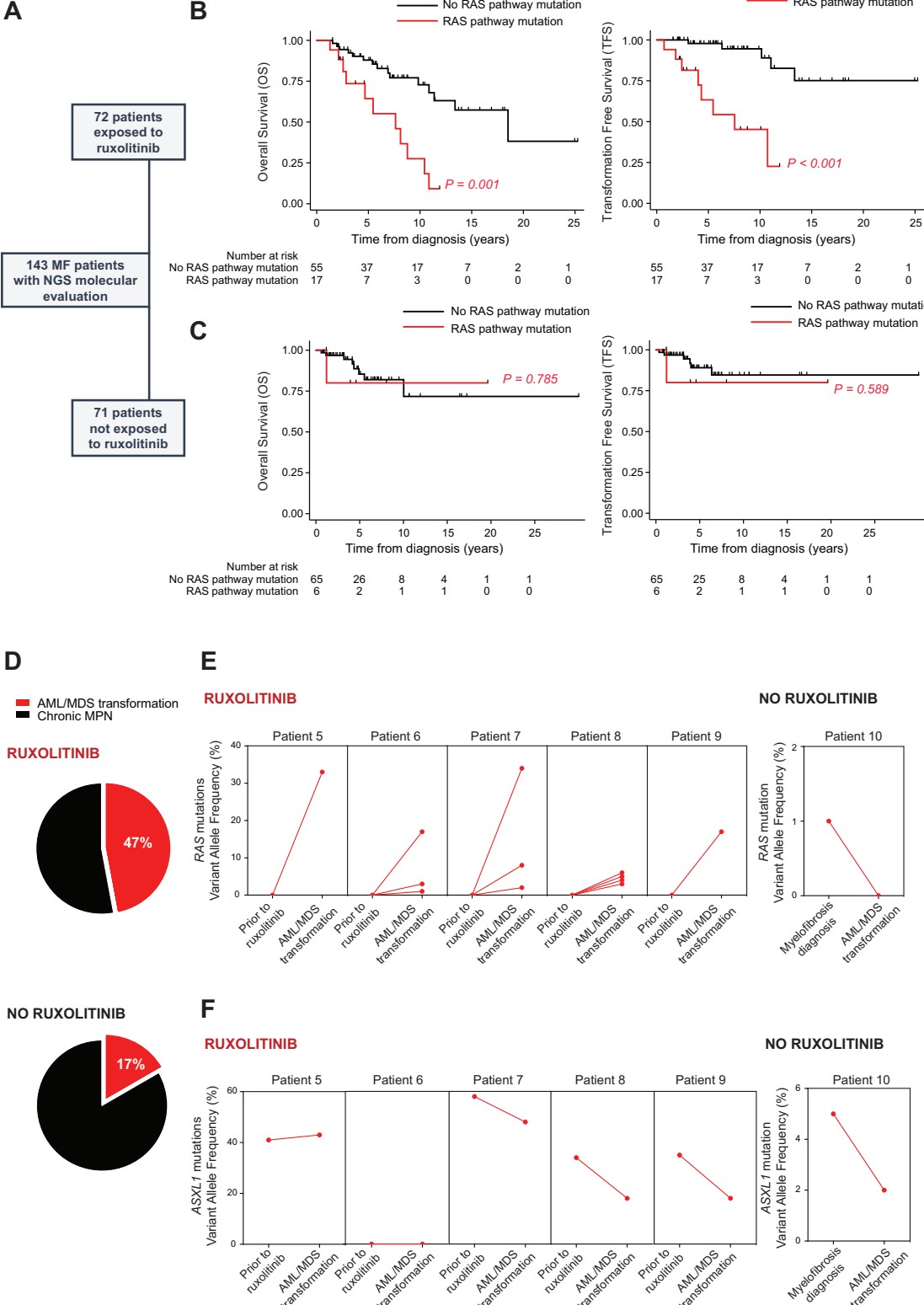

To extend our findings to the $Jak2^{V617F}$ hematopoietic cellular context, thus mimicking patients for whom RAS pathway mutations are harbored within the JAK/STAT activated MPN driver clone such as Patient#4 in Fig. 3C, we performed a second in vivo competition experiment. cKit$^+$ murine myeloid cells isolated from $Nras^{WT}$ $Jak2^{V617F}$ transgenic mice were transduced with an exogenous GFP-$Nras^{Q61K}$-expressing vector and transplanted into lethally-irradiated secondary C57BL/6 recipient mice. Ruxolitinib treatment was started 3 weeks after bone marrow transplantation (Fig. 4E). In vivo GFP monitoring through bone marrow biopsy, revealed an increased $Nras^{Q61K}$ cellular proportion in mice treated with ruxolitinib, in comparison with vehicle-treated control mice (Fig. 4F).

To further evaluate the impact of ruxolitinib on the oncogenic potential of *RAS* clones, we next sought to study its effects on the

**Fig. 2 | RAS pathway mutations adversely impact MPN patient prognosis and are associated with RAS driven leukemic transformation, only in the context of ruxolitinib treatment. A** Flow-chart depicting the patients included in the study according to their exposure to ruxolitinib treatment prior to their last NGS molecular evaluation. **B, C** Kaplan-Meier curves of overall survival (left) or transformation-free survival (right), for ruxolitinib treated (**B**) or non-ruxolitinib-treated (**C**) patients according to the presence of RAS pathway mutations. Two-sided COX proportional hazards regression analysis was used to compare the groups. P-value reported in the figure. **D** Pie Chart depicting the proportion of AML/MDS transformations among *RAS*-mutated patients, treated ($n = 8/17$) or not with ruxolitinib ($n = 1/6$). **E, F** Evolution of *RAS* (**E**) and *ASXL1* (**F**) mutations variant allele frequency (VAF) between two time points: prior to ruxolitinib initiation and at time of AML/MDS transformation, for *RAS*-mutated MPN patients treated or not with ruxolitinib. Data was available for $n = 5/8$ ruxolitinib-treated patients and $n = 1/1$ non-ruxolitinib-treated patients. Source data are provided as a Source Data file.

clonogenic potential of *Nras*$^{Q61K}$ cells. We thus performed a colony formation assay on cKit$^+$ murine cells, isolated from *Jak2*$^{WT}$ or *Jak2*$^{V617F}$ C57BL/6 mice, transduced with an empty or a *Nras*$^{Q61K}$ expressing vector. While non-*RAS* mutant cells were sensitive to ruxolitinib with a serially decreased clonogenic potential, ruxolitinib did not decrease *Jak2*$^{WT}$ *Nras*$^{Q61K}$ cells clonogenic potential, and even increased *Nras*$^{Q61K}$ cells clonogenic potential in a *Jak2*$^{V617F}$ cellular context (Fig. 4G, H).

To further evaluate the effects of ruxolitinib treatment on human *RAS*-mutated cells, we transduced HEL and UKE-1 *JAK2*$^{V617F}$ human myeloid cell lines with an empty or an *NRAS*$^{Q61K}$ encoding vector. Activation of the MAPK pathway upon exogenous *NRAS*$^{Q61K}$ expression and inhibition of the JAK/STAT pathway after ruxolitinib treatment were validated by western blot. Ruxolitinib treatment for 24 h also resulted in decreased ERK phosphorylation in a *RAS* WT context (Fig. 4I), in line with the role of JAK kinases in activating not only the STAT pathway but also parallel cell survival signals including the MAPK pathway[14]. First, we evaluated the effects of ruxolitinib treatment on each cell type through a growth inhibition experiment. While *RAS* WT cell growth was impaired after 6 days of ruxolitinib treatment, *NRAS*$^{Q61K}$ cells were resistant to JAK/STAT inhibition, in both HEL and UKE-1 cell lines (Fig. S3B). Using our set of transduced cells, we then designed an in vitro competition assay. Empty vector or *NRAS*$^{Q61K}$ expressing cells were put in competition with GFP expressing cells at a 20/80 ratio, prior to treatment with either DMSO or ruxolitinib (Fig. 4J). The proportion of GFP$^+$ cells was evaluated by flow cytometry every 12 days for a total of 24 days post-ruxolitinib treatment initiation. Ruxolitinib treatment did not strongly impact the empty vector / GFP control cells ratio. Compared to empty vector cells, we observed a decrease of untreated *NRAS*$^{Q61K}$ cellular fitness, consistent with the previously reported growth arrest of *RAS*-mutated cells through oncogene-induced senescence[27]. Interestingly, this phenotype was not only reverted by ruxolitinib treatment, but we observed a time-dependent increase of the *NRAS*$^{Q61K}$ / GFP cellular ratio under ruxolitinib treatment, suggesting that *RAS*-mutant cells are more resistant to JAK/STAT pathway inhibition than *RAS* wild-type cells. To confirm that this resistant phenotype is dependent on mutant *RAS* induced MAPK pathway activation, we exposed our *NRAS*$^{Q61K}$ / GFP cellular mix previously treated with ruxolitinib, to either DMSO, ruxolitinib or a combination of ruxolitinib and the MEK1/2 inhibitor trametinib. Trametinib treatment partially reverted the ruxolitinib-induced *NRAS*$^{Q61K}$ cellular fitness increase, suggesting that the MAPK pathway inhibition re-sensitizes *NRAS*$^{Q61K}$ cells to ruxolitinib (Fig. 4K). These results are in line with the previously described synergistic effects of JAK/STAT and MEK/ERK inhibition in MPN murine models[28].

Importantly, *Nras*$^{G12D}$, *NRAS*$^{G12V}$, and *KRAS*$^{G13D}$ HEL expressing cells were also resistant to JAK/STAT inhibition (Fig. S3C, D), and ruxolitinib exposure resulted in the positive selection of *NRAS*$^{G12V}$ and *KRAS*$^{G13D}$ HEL expressing cells (Fig. S3E), highlighting a similar impact on different *RAS* isoforms and mutations.

Additionally, to evaluate the effect of ruxolitinib on the clonal selection of *RAS* mutations within a *CALR* mutated context, we transduced Ba/F3 cells expressing exogenous human *MPL* and *CALR*$^{WT}$ or *CALR*$^{delS229}$ with an empty or a *NRAS*$^{Q61K}$ encoding vector (Fig. 4L). While

*RAS* WT cell growth was impaired after 3 days of ruxolitinib treatment, Ba/F3 *MPL-CALR*$^{WT}$-*NRAS*$^{Q61K}$ and Ba/F3 *MPL-CALR*$^{delS2}$-*NRAS*$^{Q61K}$ cells were resistant to JAK/STAT inhibition (Fig. S3F). Ruxolitinib treatment also resulted in the positive selection of Ba/F3 *MPL-CALR*$^{WT}$-*NRAS*$^{Q61K}$ and Ba/F3 *MPL-CALR*$^{delS2}$-*NRAS*$^{Q61K}$ cells in our in vitro competition assay (Fig. 4M).

Finally, to further evaluate the specificity of ruxolitinib-mediated *RAS* clonal selection in vitro, we assessed the effect of ruxolitinib exposure on *ASXL1* mutations selection. Given the controversy around the gain- or loss-of-function impact of *ASXL1* mutations in myeloid malignancies[30], we developed two complementary strategies to evaluate the impact of ruxolitinib on clonal selection of HEL cells: 1) overexpression of an *ASXL1*$^{G646Wfs*12}$ mutation (Fig. S4A–C), or 2) *ASXL1* knock-down using 2 shRNAs targeting *ASXL1* (Fig. S4D–F). In both cases, ruxolitinib exposure only mildly remodeled the fitness of the *ASXL1* clone (Fig. S4A–F), suggesting a specific effect of ruxolitinib on RAS pathway-mutated clones.

Taken together, our results argue that MAPK pathway activation in the context of RAS pathway mutations drives ruxolitinib resistance, enabling cellular escape from JAK/STAT pathway growth inhibition and a subsequent fitness advantage over *RAS* WT clones.

## JAK2 Inhibition mediates ruxolitinib's enhancement of RAS mutant cell fitness and increased clonogenic potential

As an orthogonal approach to evaluate the direct effect of ruxolitinib-induced JAK2 inhibition on *RAS* clonal selection, we next used shRNA-mediated *Jak2* suppression. First, we transduced 32D murine cells with a CRIMSON, an empty, a GFP-*Nras*$^{Q61K}$ or a GFP-*Nras*$^{Q61K}$-sh*Jak2* encoding vector. A total of 10 shRNAs targeting *Jak2* were used in a pooled fashion. *NRAS* overexpression and *Jak2* knockdown were validated by qPCR (Fig. 5A). While *Nras*$^{Q61K}$ cellular viability was decreased in comparison with non-mutant cells, *Jak2* knockdown fully reverted this effect, enabling *RAS* mutant cells to increase their proliferative capacity (Fig. 5B).

Using our set of transduced cells, we then performed an in vitro competition experiment between CRIMSON expressing cells and either empty, GFP-*Nras*$^{Q61K}$ or GFP-*Nras*$^{Q61K}$-sh*Jak2* expressing cells, at an 80/20 cellular ratio, respectively (Fig. 5C). CRIMSON and GFP markers allowed cellular proportion monitoring by flow cytometry. *Nras*$^{Q61K}$ expression reduced cellular fitness in comparison with non-*Ras* mutant cells, while GFP-*Nras*$^{Q61K}$-sh*Jak2* cells had a partially restored cellular fitness, suggesting that *Jak2* knockdown increased *Ras* mutant cells fitness (Fig. 5D).

To further validate the effect of JAK2 impairment, we performed a colony formation assay on cKit$^+$ murine cells obtained from *Jak2*$^{WT}$ or *Jak2*$^{V617F}$ transgenic mice, transduced with an empty, a *Nras*$^{Q61K}$ or a *Nras*$^{Q61K}$-sh*Jak2* expressing vector. While the colony-formation ability of *Nras*$^{Q61K}$ cells decreased over serial replating in comparison with empty vector transduced cells, JAK/STAT signaling pathway impairment through *Jak2* knock-down restored *Nras*$^{Q61K}$ cells colony-formation ability (Fig. 5E–F). Conversely, *Jak2* knock-down reduced *Nras*$^{WT}$ cells' colony-formation ability (Figs. S5A, B).

These results are consistent with the effects of ruxolitinib treatment in increasing *RAS*-mutated cell fitness in vivo and increasing their clonogenic and stemness potential in vitro.

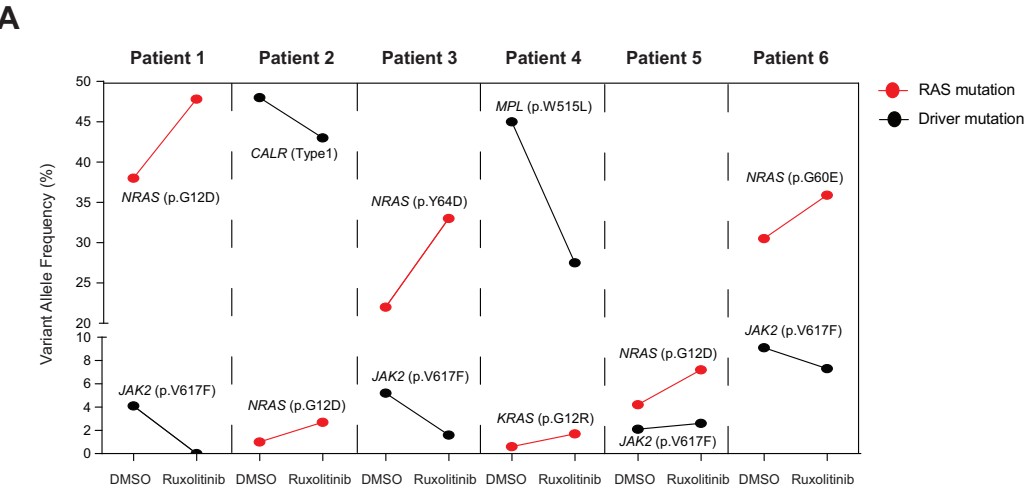

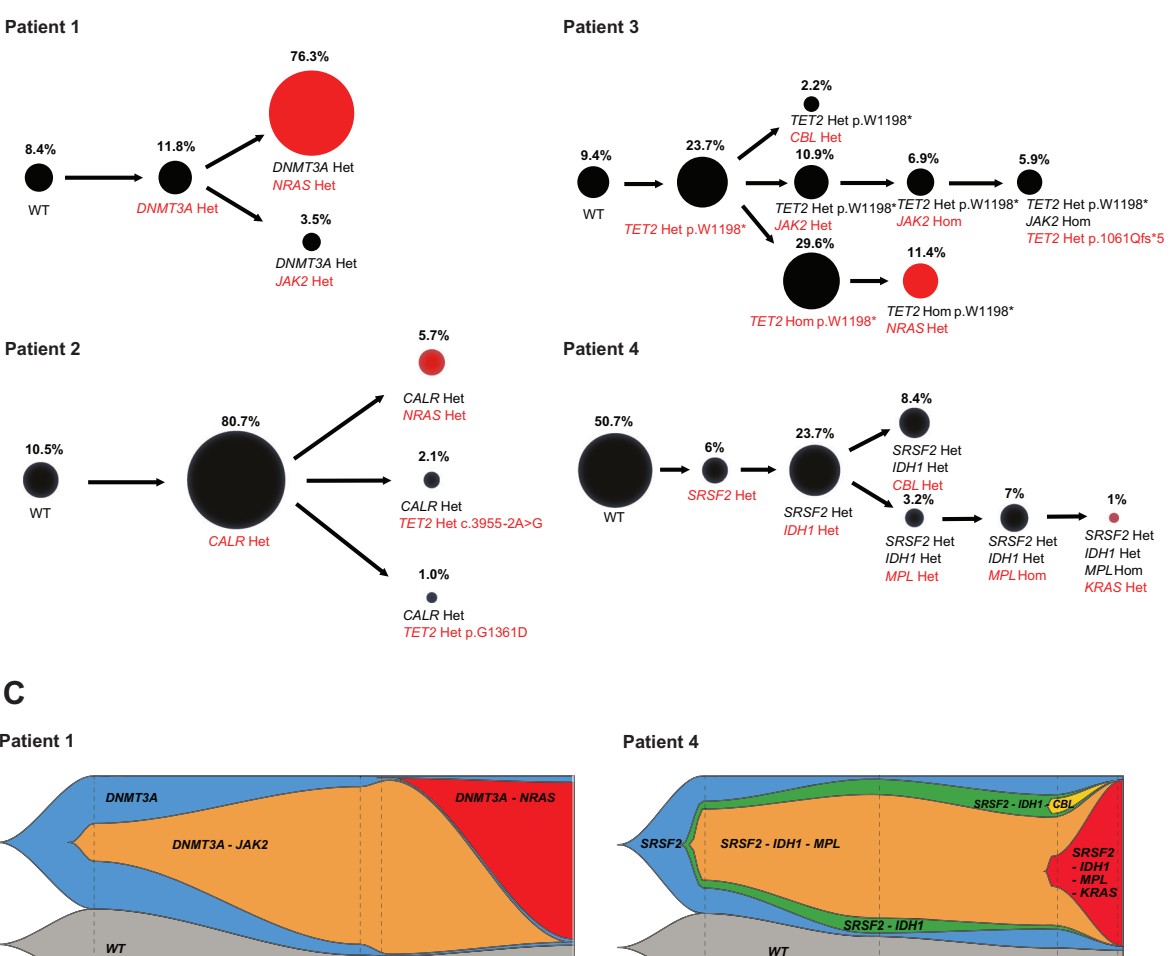

**Fig. 3 | Ruxolitinib treatment positively selects *RAS* mutant clones both in a JAK/STAT hyper-activated and wild-type (WT) context. A** Graphical representation of the allele burden of *RAS* mutations and driver mutations detected 10 days after DMSO or ruxolitinib (20 nM) in vitro treatment of CD34⁺ hematopoietic cells derived from six MPN patients harboring *RAS* mutations. **B** Clonal architecture of CD34⁺ hematopoietic cells derived from targeted single-cell DNA sequencing of samples from four patients with MPN harboring *RAS* mutations. Each

circle represents a single clone with the corresponding mutated genes below (Het = heterozygous, Hom = homozygous). Mutations acquired at each step are written in red. Red circles correspond to *RAS*-mutated clones. **C** Fish Plots depicting clonal evolution between MPN diagnosis and last molecular evaluation for two patients with MPN treated with ruxolitinib. *RAS*-mutated clones are highlighted in red. Source data are provided as a Source Data file.

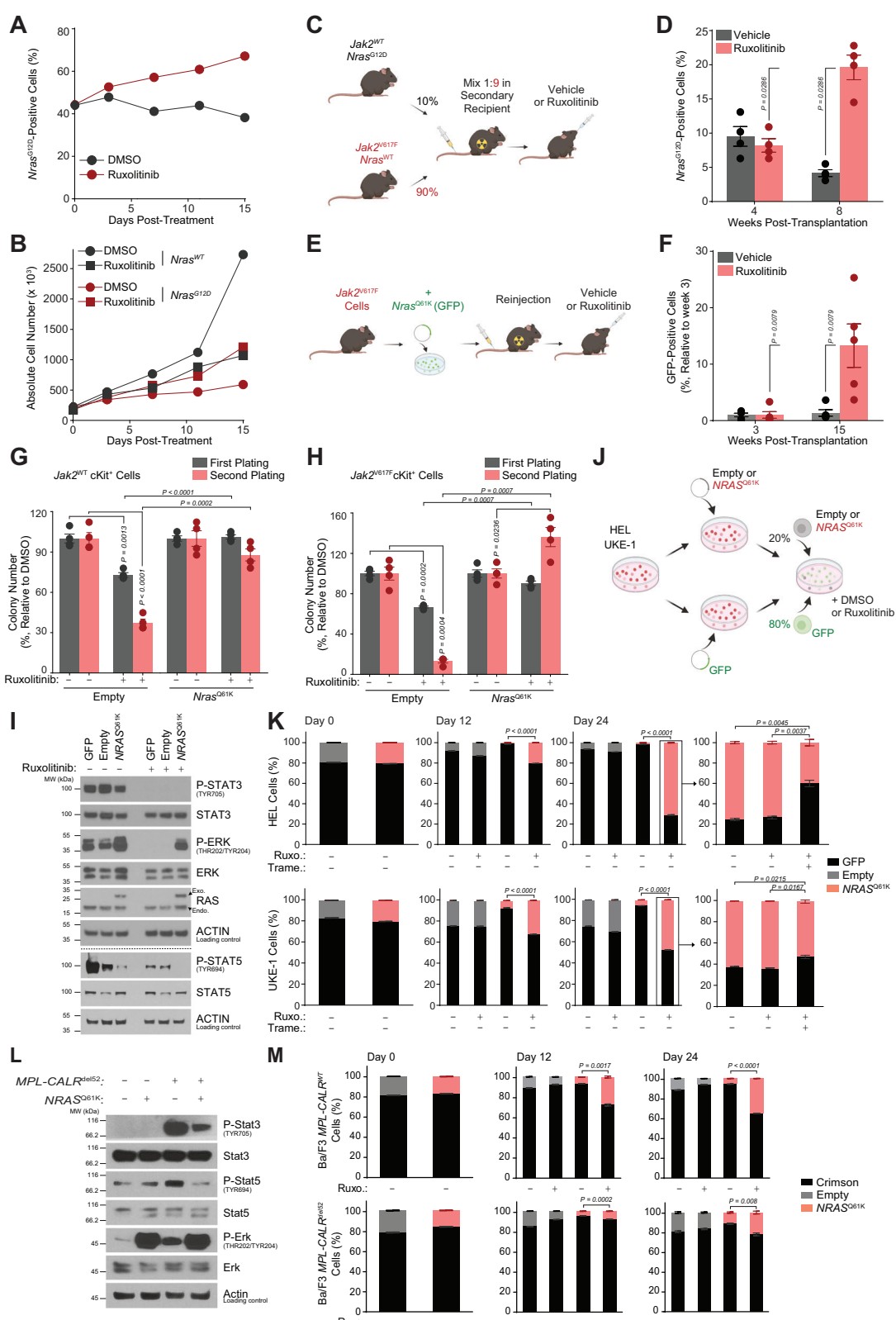

Finally, to gain more confidence in this JAK2-mediated on-target effect, we evaluated the effect of other JAK2 inhibitors harboring diverse specificities against the other JAK family members in primary AML samples. To identify specific mutational backgrounds associated with JAK2 inhibitor response in myeloid malignancies, we interrogated the Beat AML patient cohort[31]. Notably, *KRAS* and *NRAS* genetic alterations were among the top

scoring variants associated with resistance to ruxolitinib, but also to momelotinib and fedratinib, two more recently FDA approved JAK2 inhibitors, thus arguing for an on-target therapeutic class effect (Fig. 5G).

Altogether, our results confirm that ruxolitinib-mediated *RAS* clonal selection is not due to an off-target effect, but rather to a specific JAK2 mediated JAK/STAT pathway inhibition.

**Fig. 4 | MAPK activation confers fitness advantage and increased clonogenic potential to $JAK2^{V617F}$ and $JAK2^{WT}$ hematopoietic cells upon ruxolitinib.**
**A** Percentage of CD45.1/2 $Nras^{G12D}$ cells among CD45.1/2 $Nras^{G12D}$ and CD45.1 $Nras^{WT}$ Lin$^-$ murine cells (50/50 ratio) treated with DMSO or ruxolitinib (0.25 μM).
**B** Proliferation curves of $Nras^{G12D}$ or $Nras^{WT}$ Lin$^-$ cells after DMSO or ruxolitinib (0.25 μM). **A, B** Mean of $n = 2$ biological replicates. **C** In vivo $Nras^{G12D}$ competition model. Created in BioRender[59]. **D** Percentage of CD45.1/2 $Nras^{G12D}$ cells in peripheral blood of mice transplanted with 10% CD45.1/2 $Jak2^{WT}$ $Nras^{G12D}$ and 90% CD45.2 $Jak2^{V617F}$ $Nras^{WT}$ Lin$^-$ cells ($n = 4$ per group). Ruxolitinib (90 mg/kg twice daily) was started 4 weeks post-transplantation. **E** In vivo $Nras^{Q61K}$ competition model. Created in BioRender[59]. **F** Percentage of GFP$^+$ $Nras^{Q61K}$ $Jak2^{V617F}$ cells in the bone marrow of mice transplanted with $Jak2^{V617F}$ cells expressing a GFP$^+$ $Nras^{Q61K}$ vector ($n = 5$ per group). Ruxolitinib (90 mg/kg twice daily) was started 3 weeks post-transplantation. **G, H** Colony formation assay for cKit$^+$ bone marrow cells from $Jak2^{WT}$ (**G**) or $Jak2^{V617F}$ (**H**) mice expressing Empty or $Nras^{Q61K}$ vectors. Colony number after at least 6 days of DMSO or ruxolitinib (1 μM) ($n = 4$ biological replicates). **I** Western blot for the indicated proteins in HEL cells expressing GFP, Empty or $NRAS^{Q61K}$ vectors, treated 24 h with ruxolitinib (3 μM). **J** In vitro competition model. Created in BioRender[59]. **K** Percentage of GFP$^+$ and GFP$^-$_Empty or GFP$^-$_$NRAS^{Q61K}$ HEL or UKE-1 cells. 24 days after ruxolitinib (3 μM and 15 μM for HEL and UKE-1, respectively), the indicated cells were treated 3 days with ruxolitinib or ruxolitinib (Ruxo.) and trametinib (Trame.) (10 μM and 0.1 μM for HEL and UKE-1, respectively) ($n = 3$ biological replicates). **L** Western blot for the indicated proteins in Ba/F3 $MPL$-$CALR^{WT}$ and $MPL$-$CALR^{delS2}$ cells expressing Empty or $NRAS^{Q61K}$ vectors, treated 24 h with ruxolitinib (75 nM). **M** Percentage of Crimson$^+$ and Crimson$^-$_Empty or Crimson$^-$_$NRAS^{Q61K}$ Ba/F3 $MPL$-$CALR^{WT}$ and $MPL$-$CALR^{delS2}$ cells after ruxolitinib ($n = 3$ biological replicates). Statistical significance using two-tailed Mann–Whitney (**D, F**) or Welch's t-test (**H, K, M**). Experiments(**I, L**) were performed twice with similar results. Error bars represent mean ± SEM. P-values in the figure. Source data provided as Source Data file. Schemas created using BioRender.

## MAPK pathway activation is associated with JAK2 impairment resulting in $RAS$-mutated cells release from oncogene-induced senescence and increased oncogenic potential

Finally, we sought to further understand the oncological ontogeny of this seemingly paradoxical cooperation between activation of MAPK signaling and inhibition of the JAK/STAT oncogenic pathway. Indeed, exploring this oncogenic cooperation could help further decipher the enigmatic positive impact of JAK/STAT signaling pathway inhibition on $RAS$ mutant cell proliferation.

We thus explored the effects of MAPK activation on $JAK2$ expression. $NRAS^{Q61K}$ overexpression in both HEL and UKE-1 $JAK2^{V617F}$ mutated cell lines resulted in the decrease of $JAK2$ mRNA and protein levels (Fig. 6A, B). This effect was also observed in a $JAK2^{WT}$ cellular context (Fig. 5A). To further evaluate the relevance of this divergent transcriptional regulation, we evaluated whether JAK/STAT down-regulation correlated with MAPK pathway activation in the Beat AML patient cohort[31]. Patients were classified as $JAK2$ "low" or $JAK2$ "high" according to their $JAK2$ mRNA expression level (Fig. S6A) and their enrichment for three previously reported gene signatures: $JAK2$ direct targets established from $JAK2$ shRNA knock-down in the HEL cell line[32] and $JAK2^{V617F}$ homozygous or heterozygous mutants established in MPN primary patient samples[32] as evaluated by single sample Gene Set Enrichment Analysis (ssGSEA) (Fig. S6B). Next, we performed GSEA to identify differentially expressed gene sets between the $JAK2$ "low" ($n = 60$) and $JAK2$ "high" ($n = 64$) groups. We found that RAS/MAPK gene sets are enriched in the $JAK2$ "low" patient samples (Fig. 6C–E and S6C), in line with our in vitro findings in HEL and UKE-1 cell lines under ruxolitinib treatment. Interestingly, along with the RAS/MAPK enrichment in the $JAK2$ "low" group, we also observed a significant enrichment of cell-cycle related gene sets (Fig. 6C–E and S6D). Accordingly, ruxolitinib treatment enabled the cycling of $NRAS^{Q61K}$ cells as shown by propidium iodide-based cell cycle analysis. Indeed, in comparison with the DMSO control condition, a higher proportion of $NRAS^{Q61K}$ ruxolitinib-treated cells were in the G2/M phase, arguing for a released cycling checkpoint responsible for increased $RAS$ mutant cells proliferation upon JAK/STAT pathway inhibition (Fig. 6F). Additionally, an EdU incorporation assay showed increased cell cycling upon ruxolitinib exposure in an $NRAS^{Q61K}$ context in contrast to the effect of ruxolitinib on $RAS^{WT}$ cells (Fig. 6G). Given the potential for RAS oncogene-induced senescence to hamper RAS oncogenesis[27], we next investigated whether the impact of JAK/STAT pathway inhibition on $NRAS^{Q61K}$ expressing HEL and UKE-1 cell lines cycling and fitness, could potentially be due to a release from the senescent cellular state caused by the $NRAS^{Q61K}$ mutation. As expected, $NRAS^{Q61K}$ mutant cells displayed a highly senescent phenotype based on a β-galactosidase staining assay. Intriguingly, the number of $NRAS^{Q61K}$ senescent cells was decreased in the context of 6 days of ruxolitinib treatment in both HEL and UKE-1 cell lines (Fig. 6H, I). Finally, we evaluated whether extrinsically overexpressing the constitutively active form of $JAK2$ ($JAK2^{V617F}$) could impair $NRAS^{Q61K}$ mutant cell growth. Indeed, when the JAK/STAT pathway was further activated in the HEL cell line, $Nras^{Q61K}$ leukemic cells growth was impaired in comparison with $Nras^{Q61K}$ mutant cells in the absence of $JAK2^{V617F}$ exogenous overexpression (Fig. 6J and S7). Altogether, our results suggest that JAK/STAT pathway inhibition might be necessary to fully enable RAS-induced oncogenic potential, thus explaining the divergent transcriptional regulation of $JAK2$ and $RAS$.

## Discussion

Our findings strongly argue for a role for JAK2 inhibition in the selection of RAS pathway mutated clones in MPNs. This observation was suggested by Mylonas et al., who evaluated clonal evolution across 15 JAK inhibitor-treated patients with myelofibrosis[33]. However, the absence of a non-treated control group did not allow the exclusion of $RAS$ mutations acquisition as part of the natural history of myelofibrosis, thus precluding any association with ruxolitinib exposure. As in patients with AML treated with FLT3 inhibitors, the emergence of RAS pathway mutated clones in ruxolitinib-treated patients could be due to a specific resistance induced by the upregulation of RAS signaling [3]. In fact, JAK2 downstream effects are not only driven by STAT3/5 phosphorylation, but also by the activation of complementary signaling pathways such as PI3K/AKT and MAPK/ERK[14]. In line with this, both intrinsic and extrinsic MAPK activating mechanisms have been reported as limiting ruxolitinib treatment efficacy in in vitro and in vivo MPN models[28,34]. Indeed, PDGFRA remains activated under ruxolitinib, leading to ERK activation in $JAK2^{V617F}$ and $MPL^{W515L}$ MPN models[28]. Additionally, upon phosphorylation of the splicing factor YBX1 by mutated $JAK2$, YBX1 intrinsically mediates MNK1 mRNA splicing, resulting in ERK phosphorylation and survival of $JAK2$-mutated cells[34]. Combinatorial therapeutic options with MEK and ERK inhibitors have thus emerged to increase ruxolitinib's efficacy[28,34,35] (NCT04097821). In line with this, the ectopic expression of $NRAS^{G12V}$ reduced the sensitivity of $JAK2^{V617F}$-mutated cell lines to the BAD-dependent pro-apoptotic effects of ruxolitinib[36]. Complementing these findings, our results further demonstrated the ability of RAS signaling activation to confer a constitutive resistance mechanism to JAK inhibitors within the MPN driver clone but also within neighboring non-driver clones. Importantly, while our primary patient and in vitro data points towards a similar $RAS$ clonal selection across patients harboring the three MPN driver mutations, the majority of our validation experiments were conducted within a $JAK2^{V617F}$ mutational context and further experimental data is required to more precisely evaluate the interaction between $RAS$ and $CALR/MPL$ mutations.

In addition to the ruxolitinib resistance driven by the intrinsic MAPK activation, our findings suggest that JAK2 downregulation could be required in $RAS$-mutated cells to finely tune the needed MAPK

oncogene dosage, highlighting the fact that hematopoietic cells might only tolerate a certain amount of MAPK. This is reminiscent of the "Goldilocks principle" of oncogene pathways, where an ideal amount of MAPK is required for oncogenesis[37]. Accordingly, our findings show that further activation of the MAPK pathway through constitutive activation of JAK2 results in an anti-oncogenic effect, while further

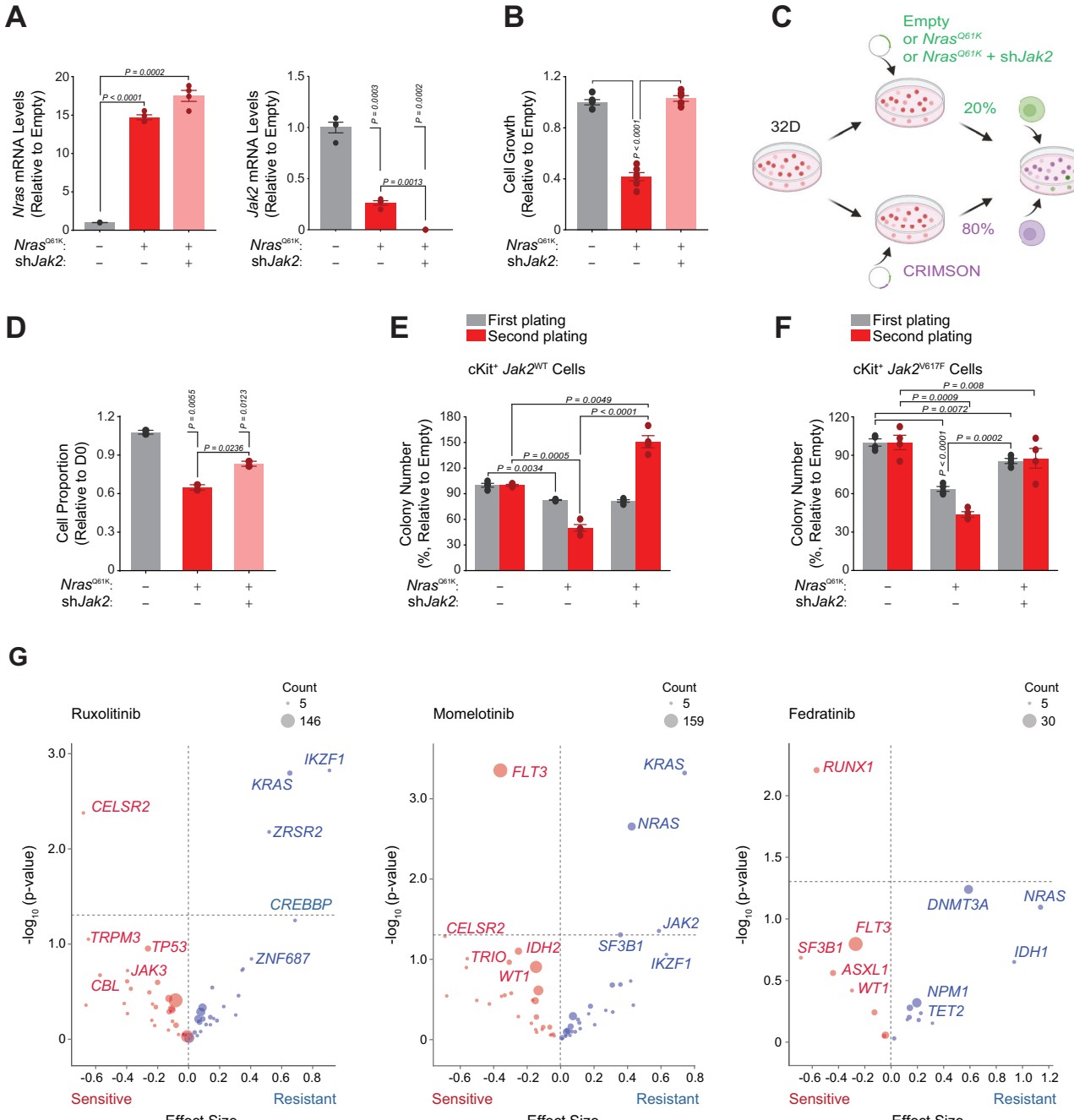

**Fig. 5 | JAK2 targeting is responsible for the effects of ruxolitinib on the fitness advantage of *NRAS* mutant cells and their increased clonogenic potential.**
**A** qRT-PCR for *Nras* and *Jak2* expression level in murine 32D cells infected with an Empty, a *Nras*^Q61K or a *Nras*^Q61K-sh*Jak2* encoding vector. Statistical significance was determined using a two-tailed Welch's t-test. Error bars represent the mean of *n* = 4 biological replicates ± SEM. **B** Growth inhibition of 32D murine cells expressing an Empty, a *Nras*^Q61K or a *Nras*^Q61K-sh*Jak2* encoding vector 6 days after GFP sorting. Statistical significance was determined using a two-tailed Welch's t-test. Error bars represent the mean of *n* = 6 biological replicates ± SEM. **C** Model of the in vitro *Nras*^Q61K-sh*Jak2* murine myeloid cells competition assay. Created in BioRender[59]. **D** Percentage of GFP⁺_CRIMSON⁻ 32D cells expressing an Empty, a *Nras*^Q61K or a *Nras*^Q61K_sh*Jak2* encoding vector 3 days after sorting. Statistical significance was determined using two-tailed Welch's t-test. Error bars represent mean of *n* = 3

biological replicates ± SEM. **E**–**F** Colony formation assay of *Jak2*^WT (**E**) or *Jak2*^V617F (**F**) C57BL/6 primary bone marrow c-Kit⁺ murine cells expressing an Empty, a *Nras*^Q61K or a *Nras*^Q61K-sh*Jak2* encoding vector at least 6 days after GFP sorting. Statistical significance was determined using a two-tailed Welch's t-test. Error bars represent the mean of *n* = 4 biological replicates ± SEM. **G** Genetic variants association with resistance to the JAK2 inhibitors ruxolitinib (*n* = 497), momelotinib (*n* = 476) and fedratinib (*n* = 93) according to the Beat AML v2 human primary patient AML dataset. Data is presented as volcano plots for gene variants' effect size (Glass) on the x-axis versus −log10(P-value) on the y-axis. Significance was set at −log10(P-value)>1.3, based on the two-tailed Student t-test with Welch's correction, with variants having a negative effect size associated with drug sensitivity (red), while those having a positive effect size associated with drug resistance (blue). P-values are reported in the figure. Source data provided as a Source Data file.

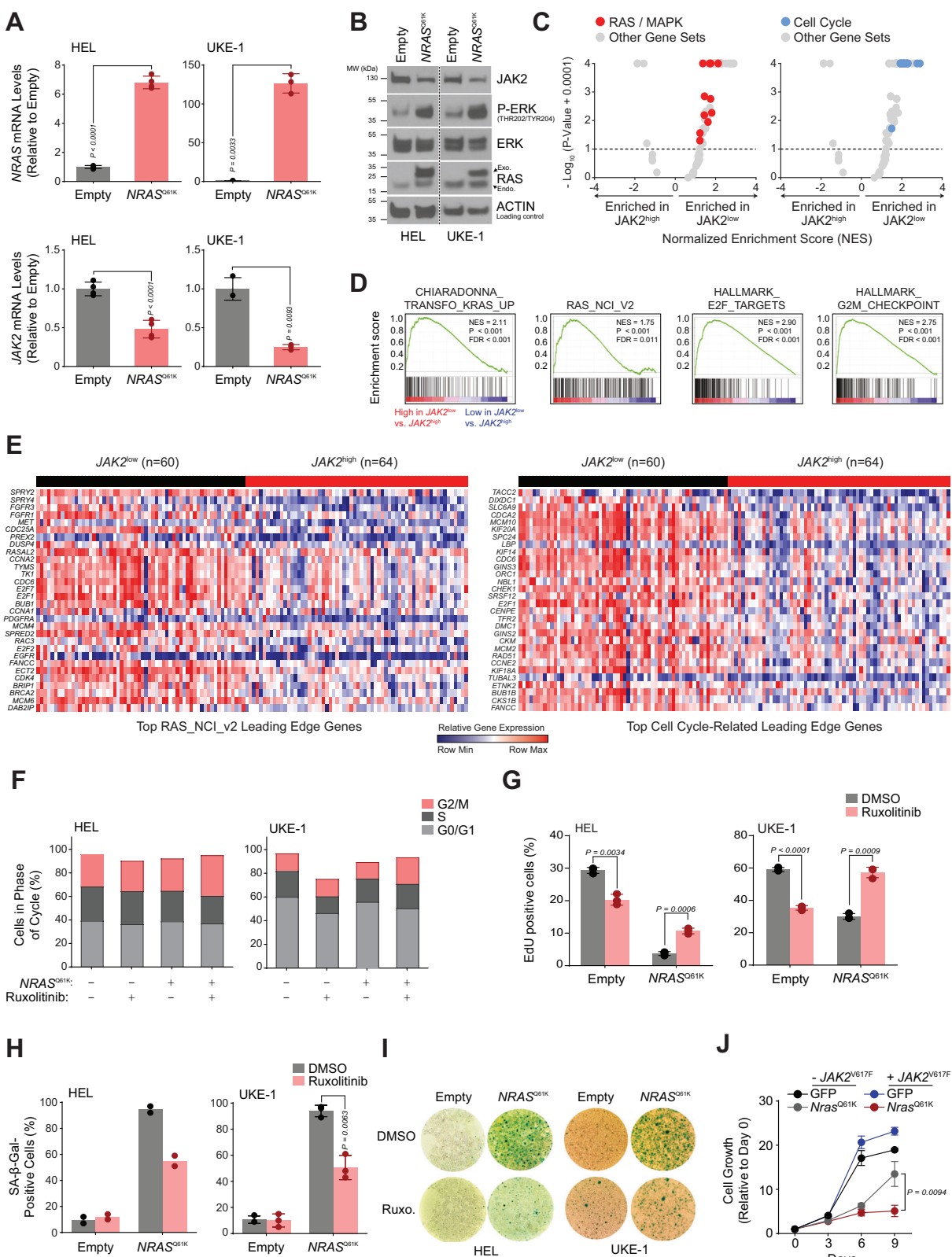

JAK2 impairment through exposure to ruxolitinib results in an increase in the oncogenic potential of *RAS*-mutated clones. In line with this concept, our data shows that *RAS* mutant clonal selection is faster in a non-JAK/STAT hyper-activated cellular context. Although an oncogene overdose principle has been documented for several oncogenic pathways, including MAPK, PI3K/AKT and WNT[38], its mechanistic underpinnings remain poorly understood. A few studies suggested increased oncogene-induced stress as the main driver of this phenomenon[39,40], in line with our findings of JAK/STAT pathway inhibition releasing *NRAS*-mutated cells from oncogene-induced senescence. Another mechanistic explanation has been recently reported in B cell leukemia, where the analysis of 1148 leukemia genomes revealed that *STAT5* and *ERK* activating oncogenic lesions are almost mutually exclusive and segregate to competing clones[41]. *STAT5* and *ERK* induce

**Fig. 6 | JAK2 downregulation increases NRAS mutated cells oncogenic potential through cellular release from oncogene-induced senescence. A** *NRAS* and *JAK2* expression level qRT-PCR in Empty or *NRAS*^Q61K HEL and UKE-1 cells. Error bars represent mean of *n* = 3 (HEL) and *n* = 4 (UKE-1) biological replicates ± SD. **B** Western blot for indicated proteins, in Empty or NRAS^Q61K HEL and UKE-1. The experiment was performed twice with similar results. **C, D** Gene Set Enrichment Analysis (GSEA) for the AML dataset Beat_AML_v2, across the collection of MSigDB_v7.4 Hallmark (50) and additional RAS/MAPK (24) and cell cycle (11) gene sets. Data represented as volcano plots (**C**) of −log10(p-value + 0.0001) versus the Normalized Enrichment Score (NES) for each gene set. NES using Kolmogorov-Smirnov enrichment test. Gene sets related to RAS/MAPK and Cell Cycle highlighted in red and blue respectively. Gray dots indicate all other. Representative GSEA plots (**D**) for gene sets related to RAS/MAPK and Cell Cycle. **E** Heatmaps depicting relative gene expression changes for leading edge genes of the top RAS/

MAPK and Cell Cycle enriched gene sets. Depleted and enriched genes are respectively in blue and red. Row-normalized data. **F** Propidium Iodide cell cycle analysis of Empty or *NRAS*^Q61K HEL and UKE-1 cells after 11 or 9 days of ruxolitinib (3 and 15 μM, respectively). Cells in sub G1 not represented. Mean of *n* = 2 biological replicates. **G** EdU incorporation of Empty or *NRAS*^Q61K HEL and UKE-1 cells after 12 days of ruxolitinib (3 and 15 μM, respectively). Error bars represent the mean of *n* = 3 biological replicates ± SEM. **H, I** Beta-galactosidase staining for Empty or *NRAS*^Q61K HEL and UKE-1 cells after 6 days of ruxolitinib (3 and 15 μM, respectively). Quantification (**H**) and representative images (**I**). Error bars represent mean of *n* = 2 (HEL) and *n* = 3 (UKE-1) biological replicates ± SD. **J** Proliferation curves of HEL cells expressing Empty, *Nras*^Q61K, *JAK2*^V617F or *Nras*^Q61K and *JAK2*^V617F vectors. Error bars represent mean ± SD of *n* = 10 (day 0), *n* = 8 (days 3, 6), *n* = 4 (day 9) biological replicates. Statistical significance using two-tailed Welch's t-test (**A, F–H, J**). P-values are reported in the figure. Source data provided as a Source Data file.

---

opposing transcriptional programs implicated in the pro-B to pre-B cellular transition. Therefore, experimental reactivation of the impaired pathway comes at the expense of the oncogenic potential of the driver oncogene. Our results complement these findings and highlight in patients the counter-intuitive oncogenic risk of impairing oncogenic pathways in specific mutational contexts.

The requirement of JAK inhibition for RAS-induced oncogenesis, as shown here in myeloid malignancies, might also be investigated in tumors with frequent RAS pathway gene mutations, such as pancreatic, skin, and lung cancers[42]. Interestingly, tumor progression has hampered the clinical development of ruxolitinib in several solid tumors (NCT01562873, NCT00638378, NCT02117479, NCT02119676, NCT02119650)[43]. Additionally, ruxolitinib treatment has been recently associated with an increased incidence of aggressive non-melanoma skin cancers[44]. In light of our results, the high dependency of cutaneous squamous cell carcinomas on MAPK pathway activation suggests a similar oncogenic mechanism potentially resulting in increased fitness of *RAS*-mutated skin cells. Targeting specific oncogenic pathways could therefore represent an extrinsic path to increased oncogenesis, reflecting the major challenge of targeting heterogeneous and adaptive diseases such as cancer. Our results warrant the search for such oncogenic interactions in order to improve our approach to the application of targeted therapies to cancer, as well as to benign disorders where patients may carry phenotypically silent somatic mutations[45].

In terms of clinical implications for the management of patients with MPN, our results suggest that ruxolitinib-induced selection of *RAS*-mutated clones might negatively impact prognosis in patients with myelofibrosis, particularly in the DIPSS intermediate-2/high risk group. *NRAS* and *KRAS* genes were identified three decades ago as important oncogenes in hematopoietic malignancies, harboring recurrent activating mutations[46–50]. Importantly, although our study highlights the impact of ruxolitinib on clonal selection of diverse *RAS* mutant cells, isoform- and mutation-specific differences in protein structure and signaling are described, and further studies are required to more precisely identify potential differences upon ruxolitinib exposure[51]. *NRAS* and *KRAS* mutations have only been recently associated with a poorer outcome and treatment resistance in MPN[25,26,52]. In our study, we confirmed the poor prognostic value of RAS pathway mutations in myelofibrosis. However, we showed that poorer OS and TFS related to these mutations are mainly observed in patients treated with ruxolitinib, while OS and TFS did not seem to be statistically impacted in patients that did not receive JAK2 inhibitor therapy. This observation suggests that the selective pressure induced by ruxolitinib could adversely impact patient outcome. Importantly, long-term findings from ruxolitinib phase 3 clinical trials did not report a higher incidence of secondary AML in the ruxolitinib-treated group[7,8,53,54]. However, the re-analysis of long-term clinical trials harbors a number of biases driven by the fact that these studies were not

designed for long-term randomized groups' comparison. Cross-over to the ruxolitinib arm, absence of longitudinal NGS molecular follow-up and the low rate of *RAS* mutations and leukemic transformation events in MPN, could have masked such observations. Given the low number of patients with RAS pathway mutations in the non-ruxolitinib treated patient group and in the DIPSS low/intermediate-1 risk categories in our study, our clinical findings need to be confirmed in larger external cohorts and across all DIPSS categories. Additionally, our results argue for an on-target JAK2 inhibition therapeutic class effect, suggesting similar findings could be observed in the clinical setting of more recently approved JAK2 inhibitors such as momelotinib, fedratinib and pacritinib. Although our results strongly suggest that JAK2 inhibitors may favor the emergence of RAS pathway mutant clones, only a subset of patients harbor such clones (24% versus 8% of ruxolitinib-treated and non-treated patients, respectively). In the context of such diseases with very limited therapeutic options, JAK2 inhibitors remain a key option in the MPN therapeutic arsenal with clear benefit for symptom management and potential survival benefits for a subset of patients.

Detailed information on ruxolitinib dose was not available for our retrospective cohort, precluding analysis of its potential impact on clinical outcomes. This should be further evaluated in future studies to inform the management of patients harboring RAS pathway mutations. Among the 8 patients with newly acquired RAS pathway mutations within the ruxolitinib-treated cohort, 6 initially responded to ruxolitinib treatment (5 clinical improvement (CI), 1 partial response (PR)), while 2 did not respond (1 stable disease (SD), 1 progressive disease (PD)). Among these patients, 5 continued on ruxolitinib treatment after identification of the *RAS* mutation while ruxolitinib was interrupted for the 3 others. Given the retrospective nature of our study, patients' treatment was not specifically adjusted after identification of RAS pathway mutations as the impact of ruxolitinib on *RAS*-mutated patient prognosis was not known at the time. Our findings argue for the implementation of sensitive molecular testing for early detection of *RAS*-mutated clones prior to JAK2 inhibitor initiation. Additionally, regular molecular follow-up followed by treatment adjustment, when possible, might also be considered. Treatment adaptation could include rationally designed combinatorial therapeutic strategies currently being evaluated in clinical trials.

In conclusion, our translational study revealed a selection of *RAS* mutations upon ruxolitinib exposure, negatively impacting clinical outcomes in patients with MPN. The enhanced oncogenic potential of *RAS* mutations after inactivation of the oncogenic JAK/STAT pathway illuminates an intriguing paradoxical oncogenic mechanism, highlights the complexity of the cancer combinatorial mutational landscape and challenges our current approach to targeted therapy use in cancer. Clinically, our results underscore the importance of monitoring adaptive responses to JAK inhibitors and considering multidrug combination approaches for patients with *RAS* mutated MPN. More

broadly, in the context of a growing number of emerging JAK inhibitors with many novel therapeutic indications for both malignant and benign diseases, our findings highlight the importance of careful monitoring for the emergence of *RAS*-driven malignancies among patients treated with JAK inhibitors.

## Methods

### Study approvals

**Patients' study.** The study was performed in accordance with the ethical guidelines of the Declaration of Helsinki, and was approved by our institutional review board (APHP Paris Nord IRB00006477, CER-2020-55). Patients provided informed consent for molecular and clinical data analysis. A unique anonymized database was established and housed in a secured system, meeting the security standards required by the protection of personal data law promulgated on 20/06/2018 in FRANCE. A diagnosis of Compliance and Security Research was carried out and approved by the data protection reference department of Saint-Louis Hospital.

**Animal study.** The French National Ethics Committee on Animal Care reviewed and approved all mouse experiments described in this study (APAFIS #34469-2021122222426491 v3). Housing conditions within Saint-Louis research institutes' animal facility satisfy the French National Ethics Committee on Animal Care requirements.

### Cell culture

**Cell lines.** 32D cell line was a kind gift of Dr Iannis Aifantis to Camille Lobry (Saint-Louis Research Institute, Paris, FRANCE). HEL cell line was purchased from the American Type Culture Collection. UKE-1 cell line was a kind gift of Dr Walter Fiedler (University Hospital Eppendorf, Hamburg, GERMANY). Ba/F3 cells expressing exogenous MPL and CALR$^{WT}$ or CALR$^{del52}$ (Ba/F3 MPL-CALR$^{WT}$ and Ba/F3 MPL-CALR$^{del52}$ cell lines) were previously published[29]. All post-MPN AML cell lines were confirmed by STR genotyping. HEL cells were maintained in RPMI 1640 (Gibco) supplemented with 1% penicillin-streptomycin (Gibco) and 10% fetal bovine serum (Sigma-Aldrich) at 37 °C with 5% CO2. UKE-1 cells were maintained in IMDM (Gibco) supplemented with 1% penicillin-streptomycin (Gibco), Hydrocortisone (1μmol/L), and 20% fetal bovine serum (Sigma-Aldrich) at 37 °C with 5% CO2. 32D cells were maintained in RPMI 1640 (Gibco) supplemented with 1% penicillin-streptomycin (Gibco), 10% fetal bovine serum (Sigma-Aldrich), and 1 ng/ml of mIL-3 (Peprotech) at 37 °C with 5% CO2. Ba/F3 cells were maintained in RPMI 1640 (Sigma-Aldrich) supplemented with 10% fetal bovine serum (Sigma-Aldrich), 1% penicillin-streptomycin (Gibco,) and 3 ng/mL of mIL-3 (Peprotech). HEK-293T cells were maintained in DMEM (Gibco) supplemented with 1% penicillin-streptomycin (Gibco) and 10% fetal bovine serum (Sigma-Aldrich).

**Primary cells.** Primary CD34$^+$ cells were magnetically sorted (EasySep Human CD34 Pos Selection Kit, Cat #17856, Stemcell Technologies) from primary MPN patients' peripheral blood mononuclear cells (PBMC) and cultured in CTS™ StemPro™ HSC Expansion Medium (Thermo Fisher) with the appropriate cocktail of cytokines: IL-3 (50 ng/L), SCF (50 ng/L), TPO (100 ng/L), IL-6 (50 ng/L), FLT3-Ligand (100 ng/L) (Peprotech). This experimental workflow was specifically developed by our team to pre-clinically study drug effects on primary CD34$^+$ cells extracted from MPN patients.

To obtain *Nras*$^{G12D}$ primary hematopoietic murine cells, total bone marrow cells were isolated from femurs and tibias of 6 to 12-week-old female *hMRP8-Nras*$^{G12D}$ mice[55] and B6.SJL-Ptprc$^a$Pepc$^b$/BoyCrl Ly5.1 congenic mice (Charles River Laboratories), by flushing from the bone cavity with ice-cold PBS using a 21 G x 1.5 needle, then passed through a 70 μM cell strainer to obtain a single cell suspension, followed by red blood cell lysis (Qiagen). Lineage-negative (Lin-) cells were then magnetically sorted (EasySep Mouse Hematopoietic Progenitor Cell Isolation, Cat #19856 A, Stemcell technologies), and maintained in StemSpan SFEM (StemCell Technologies) supplemented with 1% penicillin-streptomycin (Gibco) and 20 ng/ml mIL-3, 100 ng/ml mFLT3-ligand, and 100 ng/ml mSCF (Peprotech).

To obtain *Jak2*$^{V617F}$ and *Jak2*$^{WT}$ primary hematopoietic murine cells, total bone marrow cells were isolated from femurs, spine and tibias of 6 to 12 week-old Wild-Type C57BL/6JOlaHsd (Envigo) or Vav-cre X Floxed-*Jak2*$^{V617F}$ C57BL/6 mice[56], by bone crushing then passed through a 70 μM cell strainer to obtain a single cell suspension, followed by red blood cell lysis (Sigma-Aldrich). c-Kit$^+$ cells were then magnetically sorted (CD117 MicroBeads, mouse, Cat #130-091-224, Miltenyi Biotec) and maintained in StemSpan SFEM (StemCell Technologies) supplemented with 1% penicillin-streptomycin (Gibco) and 10 ng/ml mIL-3, 10 ng/ml IL-6, 25 ng/ml mFLT3-ligand, and 25 ng/ml mSCF (Peprotech).

### Methylcellulose colony formation assay

Eight thousand c-Kit$^+$ primary murine cells were plated in four replicates into semisolid methylcellulose medium (MethoCult M3534, StemCell Technologies) supplemented with 25 ng/ml mFLT3-ligand. After at least 6 days, the colony number was counted. The remaining cells of the same condition were then resuspended, pooled, and washed once in PBS, prior to their replating at 8000 cells or a lower number if <8000 cells per replicate were recovered.

### RNA extraction and qRT-PCR analysis

RNA was extracted using RNeasy Mini Kit (Qiagen) and retro-transcription was performed using SuperScript IV Reverse Transcriptase (Invitrogen) per manufacturer's protocols. Real-time qPCR was performed using TaqMan (Applied Biosystems) or KAPA SYBR® FAST (KAPABIOSYSTEMS) protocols provided by the supplier. Data was analyzed using the QuantStudio real time PCR instrument and software (Thermofischer).

The list of probes and primers used for real time qPCR is provided in Table S9.

### Growth measurement

To assess growth, cells were plated in 384-well cell-culture coated white plates. ATP content was then measured using CellTiter-Glo (Promega) per the manufacturer's instructions. Briefly, CellTiter-Glo reagent was added to each well and incubated at room temperature for 25 min prior to luminescence reading at the indicated time points.

### Western Blotting

Cells were lysed in Cell Signaling Lysis Buffer (Cell Signaling Technology), Halt™ Protease and Phosphatase Inhibitor Cocktail (Thermofischer), resolved by gel electrophoresis using Bolt 4%–12% Bis-Tris gels (Invitrogen), transferred to PVDF membranes (Merck), and blocked for 1 h in 3% BSA (Sigma-Aldrich). Blots were incubated with primary antibodies, followed by secondary antibodies. Bound antibodies were detected using Pierce™ ECL or SuperSignal West Pico PLUS Chemiluminescent Western Blotting Substrates (Thermofischer). Blots were developed through radiographic exposition on CL-XPosure films (Thermofisher), and scanned with a CanonScan LiDE 300 scanner. Uncropped and unprocessed scans are supplied the Source Data file.

The list of antibodies used for western blotting is provided in Table S10.

### Beta galactosidase assay

Beta galactosidase level was assessed using a Senescence β-Galactosidase Staining Kit (9860, Cell Signaling) following the manufacturer's recommendations[57].

### Flow cytometry and cell sorting

For flow cytometry analysis, cells were washed with PBS-0.5% BSA-2 mM EDTA before analysis of at least 5000 cells for each condition on

BD FACSCanto II or BD LSRFortessa instruments (BD Biosciences). Murine hematopoietic chimerism analysis was performed on the indicated days using FITC anti-mouse CD45.1 (Cat 110706, Biolegend) and PE anti-mouse CD45.2 antibodies (Cat #109808, Biolegend) staining. Murine peripheral blood cells were collected and washed with PBS-0.1% BSA-2 mM EDTA before staining for 30 min at 4 °C with the corresponding cell surface antibodies. Then, cells were lysed for 10 min with red blood cell lysis buffer (Qiagen) and washed twice with PBS-0.1%BSA-2 mM EDTA prior to FACS analysis. Data was analyzed using the Diva (Becton Dickinson) or FlowJo softwares.

For cellular sorting, cells were washed with PBS-0.5% BSA-2 mM EDTA and resuspended in PBS-0.5% BSA-2 mM EDTA, prior to flow cytometry sorting of the cellular populations expressing each fluorescent marker of interest using a BD FACSAria II instrument (BD Biosciences).

### Cell cycle analysis

For propidium iodide staining, cells were harvested at the indicated time points, washed in PBS and fixed in ice cold 70% ethanol for a minimum of 12 h. Cells were then washed in PBS, resuspended in PBS and incubated for 5 min at 4 °C with RNAse A (30 mg/ml, Sigma), prior to their incubation for 30 min in propidium iodide (1 mg/ml, Invitrogen). FACS analysis was performed on a BD FACSCanto II instrument (BD Biosciences) directly after incubation.

For the EdU assay, cells were treated with 10 µM EdU during 2 h, before being washed with PBS, fixed and permeabilized (Cytofix/Cytoperm kit #554714, BD Biosciences), at the indicated time points. Cells were then processed per manufacturers' instructions (EdU Click-iT™ Alexa Fluor™ 647, Cat C10340, ThermoFisher). FACS data acquisition and analysis were performed on a BD FACSCanto II instrument (BD Biosciences) and using FlowJo software.

### Chemicals

Ruxolitinib and Trametinib were purchased from MedChemExpress.

### Plasmids, shRNA Constructs and cell infection

pLEX-FHH-IRES-Puro (Empty Vector) and pLEX-FHH-*NRAS^{Q61K}*-IRES-Puro (*NRAS^{Q61K}* Vector) were a gift from Paul Khavari (Addgene # 120568 and # 120570). pLenti-PGK-GFP-Puro (w509-5) (GFP Vector) was a gift from Eric Campeau & Paul Kaufman (Addgene # 19070). pLenti-PGK-Hygro-DEST-w530-1 (Empty Vector) and pLenti-PGK-NRASG12V were a gift from Eric Campeau, Paul Kaufman & Daniel Haber (Addgene # 19066 and # 35632). MSCV-IRES-GFP (Empty Vector), MSCV-IRES-mCherry FP (mCherry Vector) and MSCV-IRES-GFP ASXL1 (ASXL1G646Wfs*12 Vector) were a gift from Tannishtha Reya, Dario Vignali & Anjana Rao (Addgene # 20672, # 52114 and # 81021). pcw107 (Empty Vector) and *JAK2* (V617F)-pcw107-V5 (*JAK2^{V617F}* Vector) were a gift from John Doench, Kris Wood & David Sabatini (Addgene # 62511 and # 64610).

Crimson and Nras^{Q61K} vectors were designed by cloning crimson and Nras^{Q61K} cassettes in place of the native eGFP sequence of the SGEN vector (Addgene # 111171), using the AscI and XhoI restriction sites, and by replacing the NeoR-encoding sequence by an eGFP cassette using the BspEI and SalI restriction sites. Nras^{G12D} vector was established by cloning an Nras^{G12D} cassette using XhoI and EcoRI restriction sites into a pMIG empty plasmid in which the native eGFP cassette was replaced by a Crimson encoding sequence through NcoI and PacI restriction sites. KRAS^{G13D} vector was established by substituting the native eGFP cassette of the SGEN vector (Addgene # 111171) by a KRAS^{G13D} encoding cassette, using AscI and XhoI restriction sites.

shRNA constructs targeting *Jak2* and *ASXL1* – sequences listed below – were cloned respectively into the SGEN vector (Addgene # 111171) or the LENC vector (Addgene # 111163), as XhoI–EcoRI fragments, which were generated by amplifying 97-mer oligonucleotides (Invitrogen) using 5'miRE-XhoI (TGAACTCGAGAAGGTATATTG

CTGTTGACAGTGAGCG) and 3'miRE-EcoRI (TCTCGAATTCTAGCCCCTTGAAGTCCGAGGCAGTAGGC) primers and the Vent polymerase kit (Invitrogen) with the following conditions: 50 µl reaction containing 0.05 ng oligonucleotide template, 1× Vent buffer, 0.3 mM of each dNTP, 0.8 µM of each primer, and 1.25 U Vent polymerase; cycling: 94 °C for 3 min; 35 cycles of 94 °C for 30 s, 54 °C for 30 s, and 75 °C for 20 s; 75 °C for 5 min. For our *shJak2* experiments, a total of 10 shRNAs targeting *Jak2* were used in a pooled fashion.

The list of shRNA sequences is provided in Table S11.

For virus production 18 × 10^6 HEK-293T cells were plated in 15-cm plates and transfected with 13.5 µg DNA for each lentiviral vector and with 11µg of PAX2 and 5.5 µg of VSVG packaging vectors. Viral supernatants were harvested 72 h later and filtered using 0.45-µm filters.

Human and murine cell lines were spin-infected for 2 h at 37 °C with 3 ml of lentiviral supernatant, 25 mM HEPES (Sigma-Aldrich) and 8 µg/ml polybrene (Sigma-Aldrich), and selected 48 to 72 h later with 1 µg/ml or 2 µg/ml of puromycin (Invivogen) for UKE-1 and HEL cells respectively, or sorted according to the expressed fluorescent marker.

Primary murine cells were spin-infected for 2 h at 37 °C with 4 ml lentiviral supernatant, 25 mM HEPES (Sigma-Aldrich) and 8 µg/ml polybrene (Sigma-Aldrich), on RetroNectin coated plates following manufacturer's recommendations (Takara Bio).

### In vivo transplantation

Lethally irradiated (9 Gy) 8-week-old C57BL/6 female mice were tail-vein injected with a total of 2×10^6 Lin⁻ *Jak2^{WT} Nras^{G12D}* and *Jak2^{V617F} Nras^{WT}* hematopoietic cells obtained as described above, at a 10/90 cellular ratio, respectively. Four weeks post-transplantation, mice were randomized to be treated with Ruxolitinib (90 mg/kg, oral gavage, twice daily) or vehicle (0.5% methylcellulose (w/v) and 0.1% Tween 80) for 28 days. Peripheral blood mononuclear cells were collected by submandibular bleeding at the indicated time points.

Lethally irradiated (2 ×4.5 Gy) 6-week-old C57BL/6 male mice were tail-vein injected with 1×10^6 *Jak2^{V617F}* c-Kit⁺ hematopoietic cells obtained as described above and lentivirally transduced with the SGEN-eGFP-*Nras^{Q61K}* overexpressing vector. 0.3×10^6 total murine wild-type support bone marrow was concomitantly injected in each mouse. Three weeks post-transplantation, mice were randomized to be treated with Ruxolitinib (90 mg/kg, oral gavage, twice daily) or vehicle (5% dimethyl acetamide, 0.5% methylcellulose), using a 4 weeks ON / 2 weeks OFF treatment schedule for up to 12 weeks total. Bone marrow cells were collected by bone marrow biopsy performed on mice femurs, at the indicated time points.

Mice were euthanized if they exhibited signs of moribund condition, which included hunched posture, reduced activity, labored breathing and weight loss ( > 15%) as humane endpoint.

### Patients' population

A total of 143 consecutive patients from whom a next generation sequencing (NGS) molecular analysis was performed at diagnosis and/or during follow-up were diagnosed with primary or secondary myelofibrosis (MF) according to WHO criteria and followed in Saint-Louis hospital between January 2011 and November 2019. Among these patients, 73 patients had a longitudinal molecular follow-up defined as at least 2 NGS evaluations, and 72 patients received Ruxolitinib treatment prior to their last NGS molecular evaluation.

Clinical and molecular characteristics at time of diagnosis and during follow-up were collected from medical charts and electronic medical records. Prognostic scores were evaluated both at diagnosis (IPSS) and at time of molecular evaluation (DIPSS).

### Patient samples NGS molecular data acquisition

We used a capture-based custom next-generation sequencing (NGS) panel (Sophia Genetics) targeting 36 myeloid genes (*ABL1*; *ASXL*; *BRAF*; *CALR*; *CBL*; *CCND2*; *CEBPA*; *CSF3R*; *CUX1*; *DNMT3A*; *ETNK1*; *ETV6*; *EZH2*;

*FLT3; HRAS; IDH1; IDH2; IKZF1; JAK2; KIT; KRAS; MPL; NFE2; NPM1; NRAS; PTPN11; RUNX1; SETBP1; SF3B1; SH2B3; SRSF2; TET2; TP53; U2AF1; WT1; ZRSR2)* (Table S12). Libraries were prepared using 200 ng of DNA extracted from whole blood or CD34+ patient-derived cells (Qiagen). Sequencing was then performed on a MiSeq instrument (Illumina). Bioinformatics were carried out at Sophia Genetics (Switzerland) using the SOPHIA DDM software and significant variants were retained with a sensitivity of 1%. Cbioportal oncoprinter was used to generate oncoprint plots (https://www.cbioportal.org/oncoprinter). Fishplots depicting phylogenetic trajectories were inferred from single-cell DNA sequencing combined with longitudinal NGS analysis, and generated using "fishplot" R package (version 4.1.1 for Mac). Mutations detected by NGS and not detected in our single cell DNA analysis (either absent at time of single-cell sequencing or not included in our single-cell DNA sequencing panel) were not included in the Fishplots representation to avoid incorrect clonal inference.

### Patient samples single-cell DNA sequencing

Primary CD34+ cells from primary MPN patients' peripheral blood were resuspended in Tapestri cell buffer (Mission Bio) and quantified using an automatic cell counter (Biorad). Single cells (3000-4000 cells per μL, >80% viable) were encapsulated using a Tapestri microfluidics cartridge (Mission Bio), lysed, and barcoded. Barcoded samples were then subjected to targeted polymerase chain reaction (PCR) amplification of a custom 96 amplicons covering 17 genes known to be mutated in MPN (*ASXL1; CALR; CBL; DNMT3A; EZH2; IDH1; IDH2, JAK2; KRAS; MPL; NFE2; NRAS; SF3B1; SRSF2; TET2; TP53; U2AF1*). PCR products were removed from individual droplets, purified with Ampure XP beads (Beckman Coulter), and used as a template for PCR to incorporate Illumina i5/i7 indices. PCR products were purified a second time, quantified via an Agilent Bioanalyzer, and pooled to be sequenced. Library pools were sequenced on using NextSeq 550 instrument (Illumina). Fastq files were processed using the Tapestri Pipeline for cell calling and using Genome Analysis Toolkit 4/haplotype caller for genotyping. The Tapestri Insights software was used to further filter variants, and samples were included if they harbored 3 or more protein-encoding, non-synonymous/insertion/deletion variants and >1000 cells with definitive genotype for all protein-coding variants within the sample. We next sought to define genetic clones, which we identified as cells that possessed identical genotype calls for the protein-encoding variants of interest. Importantly, almost 95% of the panel amplicons in each sample had sufficient coverage to annotate variants. The estimated median allele dropout rate was 7.54% (IQR, 5.6% to 9.3%). From the variants annotated for each sample, we first removed those with low quality ( < 30% in the Tapestri Insights software) and low frequency ( < 0.5% cells). As a result, we detected a total of 17 different variants passing the pre-filtering step across the 4 patients, with a median of 4.5 variants per patient (range from 3.0–5.0). All variants detected in bulk whole-blood NGS analysis below the 0.5% threshold were not considered in the single-cell analysis.

### Gene variants and drug effects associations

The Beat AML v2 cohort[31] was interrogated to evaluate single-nucleotide variants (silent variants excluded) association with resistance to the JAK2 inhibitors ruxolitinib (497 samples with genotypes), momelotinib (476 samples with genotypes) and fedratinib (93 samples with genotypes). Drug AUC data is available from https://biodev.github.io/BeatAML2/ and drug effect visualization from the Vizome platform (http://vizome.org/). Data is presented as volcano plots for gene variants effect size (Glass) on the x-axis vs. −log10(P-value) on the y-axis. Significance was set at − log10(P-value)>1.3, with variants having a negative effect size associated with drug sensitivity, while those having a positive effect size associated with drug resistance.

### Gene Set enrichment Analysis

To accurately define patients harboring low vs. high JAK/STAT pathway activation among the 707 Beat AML v2 cohort patients for which bulk RNA-seq expression data is publicly available[31], we classified patients based both on their level of *JAK2* expression, and on their single sample GSEA scores for JAK2 direct targets and *JAK2*^V617F mutant gene signatures[32]. The functional associations of the molecular phenotypes were explored with the single sample GSEA (ssGSEA) method[58] based on the Bioconductor GSVA v1.40.1 implementation. ssGSEA is an extension of Gene Set Enrichment Analysis that calculates separate enrichment scores for each pairing of a sample and gene set. Each ssGSEA enrichment score represents the degree to which the genes in a particular gene set have coordinately increased or decreased expression within a sample. *JAK2* "low" and "high" groups were defined based on the consensus conditions between i) *JAK2* expression z-scores across the full cohort (<-0.5 or >0.5 respectively), and ii) at least two of the three ssGSEA scores for shJAK2 vs. GFP[32], *JAK2*^V617F homozygous vs. Normal and/or *JAK2*^V617F heterozygous vs. Normal gene signatures[32] ( < -0.5 or >0.5 respectively). Beat AML v2 *JAK2* "low" (*n* = 60) and "high" (*n* = 64) groups definition was validated for *JAK2* expression (log2(TPM + 1)) and association with *JAK2* gene target and mutated gene set signatures through Genome-wide Gene Set Enrichment Analysis for enrichment in expression changes. The Gene Set enrichment analysis (GSEA) v4.2.0 software was used to identify functional associations of molecular phenotypes with gene sets included in the MSigDB v7.4 database and the National Cancer Institute's RAS Initiative (https://www.cancer.gov/research/key-initiatives/ras) (Supplementary Data 2). The goal of GSEA was to identify the gene sets that are distributed at the top or at the bottom of the ranked list of genes based on the Kolmogorov-Smirnov enrichment test. Gene sets with absolute Normalized Enrichment Score (NES) ≥ 1.3, a nominal p-value ≤ 0.05 and an FDR ≤ 0.25 for the Kolmogorov-Smirnov test were considered significant hits. The results were visualized on volcano plots for the normalized enrichment score (NES) vs. -log10(p-value) and on GSEA plots. Heatmaps depicting relative gene expression changes for leading edge genes for the top enriched gene sets were also represented.

### Statistics

**In vitro and in vivo statistical analysis.** Statistical analysis was performed using PRISM 8.0.1 (GraphPad). Data were analyzed using a nonparametric Mann-Whitney test (with the assumption of no Gaussian distribution of the group) or parametric Welch's correction (with the assumption that both groups of data are sampled from Gaussian populations), unless otherwise specified, and the threshold of significance (α) was set at 0.05.

**Patients' cohort statistical analysis.** Continuous variables are reported as medians and interquartile ranges (IQR), while categorical variables are reported as numbers and proportions. Comparison of continuous and categorical variables between subgroups was performed by Mann-Whitney test and Fisher's exact test, respectively.

RAS pathway mutations cumulative hazard was estimated by the Nelson-Aalen method and compared by cause-specific hazard Cox model. Time to mutation was calculated from the date of first available NGS evaluation to the last available NGS molecular assessment. Ruxolitinib treatment was considered a time-dependent covariate. OS and transformation-free survival (TFS) were estimated using the Kaplan-Meier method. OS was measured from the date of MF diagnosis until death, censoring patients still alive at the date of last follow-up and patients undergoing Hematopoietic Stem Cell Transplantation (HSCT) at the time of transplantation. TFS was measured from the date of MF diagnosis until the date of the transformation (AML or MDS), censoring patients still alive without transformation at the date of last follow-up and patients undergoing HSCT at the time of transplantation. A

total of eight patients (5/71 in the non-ruxolitinib-treated group and 3/72 in the ruxolitinib-treated group) had HSCT.

Univariate and multivariate analyses assessing the impact of categorical and continuous variables on "RAS pathway" mutations acquisition, OS or TFS were performed using Cox regression models. The proportional hazard assumption was validated. All significant variables ($p < 0.05$) were included in the multivariate analysis. Hazard ratios (HR) are given with 95% confidence interval (CI). Statistical analyses were performed using the STATA software (STATA 15.1 Corporation, College Station, TX).

### Reporting summary

Further information on research design is available in the Nature Portfolio Reporting Summary linked to this article.

## Data availability

Human single cell DNA sequencing data generated in this study are available at the Sequence Read Archive (SRA) repository of the National Center for Biotechnology Information (NCBI). The accession number for these SRA data is PRJNA1222460 and data can be accessed through the following link: https://www.ncbi.nlm.nih.gov/sra/PRJNA1222460. Supplementary information, including Supplementary Figs. and legends, Supplementary Tables, Supplementary Data and Source data are provided with this paper. Sources for reagents and cells are indicated in the Materials and Methods section. No custom code was generated in the course of this study. Source data are provided with this paper.

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

## Acknowledgements

The authors thank the clinical care team of the Comprehensive Myeloproliferative neoplasms Center for samples and data collection, and the staff of the cellular biology laboratory for excellent technical assistance. We are indebted to Veronique Montcuquet, Niclas Setterblad, Christelle Doliger, and Claire Maillard from the Saint-Louis Research Institute Core Facility for their technical support. The authors also thank the French Intergroup for Myeloproliferative neoplasms (FIM) for insightful discussions. This work was supported by an "Association Laurette Fugain" (to J.J.K and L.B.), a "Fédération Leucémie Espoir" (to J.J.K and L.B.), a "Fondation ARC pour la recherche sur le cancer" (to J.J.K.), an "INCa Prev-Bio" (to J.J.K and L.B.), a "CCA-INSERM-Bettencourt" (to L.B.,) and an "ATIP-Avenir / Ligue National Contre le Cancer" (to L.B.) funding. L.B. is supported by the ERC Starting program (101117339) and an Emergence Ville de Paris grant. N.K. is supported by the "Fondation pour la Recherche Médicale - FRM" / "Fondation Capucine". K.S. was supported by the National Cancer Institute R35 CA210030. This work was supported by the Groupe Francophone des Myelodysplasies (to R.A.P.). P.G. was supported by the Chinese Scholarship Council (CSC N°201706180057). A.P. is supported by the ERC Starting and Consolidator programs (758848 and 101088563). L.B., C.L., and A.P. are supported by the SIRIC InsiTu program (INCa-DGOS-INSERM-ITMO Cancer_18008) and the IHU France 2030 "Leukemia Institute Paris Saint-Louis" (ANR-23-IAHU-0005).

## Author contributions

N.K., N.M., and B.R.: formal analysis, validation, investigation, visualization, methodology, writing original draft. G.A.: conceptualization, methodology, investigation, formal analysis, visualization. H.P., E.V., R.D.O., R.M., C.C., and N.G.: investigation, methodology, resources. F.G.: conceptualization, methodology. L.P.Z.: visualization. S.Ga., P.G., B.M., and F.L.: methodology, investigation. J.S.D., N.P., W.V., E.R., R.A.P., C.M., I.P., and S.Gi.: resources, methodology. C.L. and K.S.: conceptualization, investigation, methodology. A.P.: conceptualization, validation, investigation, methodology, visualization. J.J.K. and B.C.: conceptualization, formal analysis, methodology, resources, supervision, funding acquisition, visualization, writing original draft. L.B.: conceptualization, investigation, formal analysis, methodology, resources, supervision, funding acquisition, visualization, writing original draft, project administration. All co-authors reviewed, edited and critically discussed the manuscript.

## Competing interests

L.B. received research funding from Gilead and Pfizer for unrelated projects, and personal fees from BMS, Novartis and GSK outside of the submitted work. N.G. received personal fees from Novartis, Abbvie and Astra Zeneca outside of the submitted work. K.S. is on the SAB and has stock options in Auron Therapeutics and received grant funding from Novartis and KronosBio on topics unrelated to this manuscript. The remaining authors declare no competing interests.
