## [Transparent Peer Review file · Nature Communications]

JAK2 inhibition mediates clonal selection of RAS pathway mutations in myeloproliferative neoplasms

Corresponding Author: Dr Lina Benajiba

Version 0:

Reviewer comments:

Reviewer #1

(Remarks to the Author)

In their manuscript entitled "JAK2 inhibition mediates clonal selection of RAS pathway mutations in myeloproliferative neoplasms", Maslah et al. explore the impact of RAS mutations in the setting of ruxolitinib treatment for MPN. Using various levels of evidence (patient data, patient derived single cell colonies, murine experiments, cell culture experiments), they uncover an unexpected mechanism of selection of RAS mutated clones induced by JAK2 inhibition. The finding itself is unexpected, interesting and worth reporting. I appreciate the elaborate, multi-dimensional approach including clinical and pre-clinical data. However, in its current version, the manuscript has various weaknesses. In particular, I feel that some conclusions are premature and warrant broader investigation.

Major points:

- 1) The introduction is well written but information can be shortened to a maximum of approx. 1.5 pages
- 2) Most problematic about comparing ruxo-treated vs. untreated patients is that they are clinically different cohorts with different risk profiles. As shown in table S1, diseases in the non-treated cohort are in earlier clinical stages. Comparisons between these groups should therefore be avoided, in particular with regard to leukemic transformation and OS. Therefore, I would suggest to perform analyses as follows: stratification according to DIPSS at molecular evaluation: group 1: low/intermediate-1 group 2: intermediate-2/high. Further analysis according to RAS mutational status and ruxo yes/no.
- 3) On a related note, as only 6 patients in the untreated cohort have a RAS mutation, a more careful wording must be chosen when describing potential impact on TFS and OS. This is particularly true for the conclusion in the discussion (p.19). Increased leukemic transformation was for example not described in the long-term data of the COMFORT II trial. Please comment.
- 4) What was the mean (max.-min.) treatment duration with ruxolitinib until transformation?
- 5) Did patients undergo allogeneic stem cell transplantation and if so, how was this data dealt with in the OS/TFS analysis?
- 6) A general problem of the manuscript is that all types of NRAS/KRAS/HRAS mutations are set equal. This confers to the patient data as well as to the preclinical part.
 - The manuscript provides no information on the individual mutations. This could be addressed by a lollipop plot, for example. Were the mutations only hotspot mutations or were there also non-hotspot mutations found?
 - Please explain the choice of RAS mutations analyzed in the competition experiments (Figure 4). Why was NrasG12D tested in the first part of the co-cultures whereas NrasQ61K vectors were used in the second part? Was this due to a scientific rationale or due to the availability of the models?Though it is not feasible to analyze all types of NRAS/Nras/KRAS/Kras mutations in the murine setting, the authors should at least discuss this limitation of their study. Though the biological impact of different RAS-mutations seems to point into the same direction, differences between the different hotspot mutations and isoforms are described (e.g. different effects on downstream signaling/interactions). However, in the cell culture setting (experiments depicted in Figure 6), testing at least a second NRAS hotspot mutation or KRAS mutation would greatly increase the meaningfulness and reproducibility of the study.
- 7) All experiments were performed in the setting of JAK2 V617F, the interaction between CALR and MPL was not explored. This limitation should be discussed.

8) Role of ASXL1: Figure 1B shows a relevant increase in the number of ASXL1 mutations in ruxo-treated patients and Figure 2F shows that 7/8 transformed cases had an ASXL1 mutation. The role of this mutation is not investigated or discussed in the manuscript, though when looking at the pie charts, it also appears to be positively selected. Did VAF increase during transformation?

Minor points:

9) p.2: I think it is not shown that the RAS mutations are acquired by ruxo therapy. Probably, the term “clonal outgrowth” is more accurate.

10) Table S1: Information on missing data should be included (i.e. data lacking for IPSS, DIPSS, etc.).

11) Did the NGS panel only cover RAS mutational hotspots or the whole gene? Please provide the respective information, preferably for all genes investigated.

12) Include results of NGS analysis (mutation, VAF, etc.) for each patient (table in supplement)

Reviewer #2

(Remarks to the Author)

This is a very interesting study, in which the authors examined the effect of ruxolitinib on the clonal evolution of MPN, specifically myelofibrosis. Single-cell DNA sequencing combined with ex vivo treatment of RAS mutated CD34+ primary cells, demonstrated that ruxolitinib induced RAS clonal selection. Such clonal selection was associated with decreased transformation-free and overall survival in patients treated with ruxolitinib. They also found that MAPK pathway activation was associated with JAK2 downregulation which resulted in enhanced oncogenic potential of RAS mutations. This work has important implications on our overall understanding of the mechanisms of leukemogenesis and may prove to have significant impact on the clinical management of MPNs. In general, the studies have been carefully done and the results are detailed, clear and convincing. Some minor suggestions are summarized below.

1. Perhaps the authors should include a statement about cell line authentication for the MPN derived cell lines.
2. In page 18, first sentence, the authors most likely meant to write “ineffective” instead of “effective”.
3. Deposition and accessibility of the sequencing data from patients. This will obviously be important for future investigations. It was not clear in the text how these will be made available and this should be included.

Reviewer #3

(Remarks to the Author)

This is an important, comprehensive and well-written study that reveals a potential mechanism underlying the development of ruxolitinib resistance in MPN patients. Maslah et al. demonstrate that ruxolitinib treatment induces selection of RAS mutant clones in both wild type cells and MPN cells with constitutive JAK/STAT activation. Interestingly, the authors show that the mechanism underlying this phenomenon relates to MAPK-mediated downregulation of JAK2 expression such that JAK/STAT activation is “fine-tuned” to support MAPK activation dosage. Overall, the findings are well supported, and this study represents an important advancement in the MPN field.

I have only minor suggestions that I believe will strengthen an already strong study:

1. The findings in Figure 3 are interesting, but would be more robust with the addition of more MPN patients (8-10 would be ideal).
2. The cell line and mouse assays focus on Jak2 V617F – these findings would be greatly strengthened by the addition of models for mutant CALR and MPL, to show this is generally true across driver oncogenes that hyperactivate JAK/STAT signaling.

Reviewer #4

(Remarks to the Author)

Maslah and colleagues have investigated the role of RAS pathway mutations in myelofibrosis patients treated with ruxolitinib (rux) or not. Their key finding is that rux favor the outgrowth of RAS-mutated clones, and that rux impairs the outcome of patients with RAS pathway mutations. A series of in vitro and in vivo competition experiments show that reduced JAK/STAT signaling, either by rux or by JAK2 knockdown, promotes mutant RAS-mediated cellular fitness and oncogenesis, apparently by partially blocking the senescence induced by mutant RAS. This paper has a number of interesting aspects, but also several profound weaknesses, particularly with respect to novelty and the COX model.

Major concerns

1. Novelty is limited. RAS pathway activating mutations have been described as resistance drivers in MPN and in FLT3-ITD-positive AML, IDH1/2-mutated AML. The mutual exclusivity of STAT and RAS/MAPK signaling has been reported for ALL by the Mueschen lab. All these are acknowledged by the authors, but the fact remains. One would like to see mechanistic studies that go beyond the descriptive level. For example, is the pro-oncogenic effect of inhibiting JAK/STAT mediated by

reducing the specific MAPK output emanating from JAK/STAT or by blocking a set of pSTAT3 or pSTAT5 target genes?
2. There are concerns about the COX model. It does not make sense that HMR mutations are significantly associated with OS but not TFS, suggesting that the model is unstable. What were the causes of death in the non-rux group? How were these patients treated? The individual components of the IPSS should be included as well as cytogenetics. Was the year of diagnosis different between the non-rux and rux cohorts? It is unclear why OS and TFS as not evaluated for the entire cohort in the first place, with RAS mutations and ruxolitinib included as variables, then analyzed for significant interactions. Could propensity matching be used to reduce the imbalances between the rux and non-rux groups? Validation of the results in an independent cohort is critical.

3. More details on treatment and response should be provided: (i) Please comment on specific aspects of ruxolitinib therapy such as treatment dose, duration of exposure, treatment interruptions and whether they were associated with clinical endpoints (not only survival, but also others as mentioned below). (ii) comment on whether patients initially had a clinical response to ruxolitinib and subsequently lost response upon development of RAS pathway alteration vs. they never responded to treatment even before development of RAS pathway alteration. (iii) Is there information available on subsequent therapies patients received upon development of RAS alterations (continued vs. stopped ruxolitinib, switched or added other therapies) as they could influence overall survival and transformation free survival.

4. NGS: Molecular follow-up is available only for half of the cohort. I could not find the list of genes included in the panel. Does it include PTPN11? BRAF? What is the sensitivity of the assay? Since the data are very limited for some of the analysis, it is unclear what we can conclude from it – e.g. the molecular landscape of AML (N=1 for non-rux; n = 7 for rux).

5. In the experiment shown in F4C, it is unclear why Jak2WT/NrasG12D was competed against Jak2V617F/NrasWT and not Jak2V617F/NrasWT vs Jak2V617F/NrasG12D. The system represents one equation with 2 variables and cannot be solved. This is essentially what was done in F4E/F.

6. It would strengthen the manuscript if the authors included a control consisting of an unrelated oncogene, e.g. mutant ASXL1, and showed that ruxolitinib does not select for cells expressing this variant.

7. Some of the observations are entirely expected. For instance, on p.11, F4I: “Interestingly, as opposed to the RAS WT cellular context, persistent ERK phosphorylation was observed upon ruxolitinib exposure in a RAS mutated context (Figure 4I).” Along the same lines, there is a lot of unnecessary redundancy, particularly in the discussion which should be shortened considerably.

8. I disagree with the interpretation of the cell cycle data (F6C-E). First, the subG1 cells must be removed – these are dead cells. Second, increased cell cycle entry should be manifest by more cells in S not in G2/M.

Minor concerns

1. The sentence on p.10 “Given the clinical impact of ruxolitinib-selected RAS mutations in our MPN patient cohort, we next sought to study its effects on the clonogenic potential of NrasQ61K cells, as a surrogate for their AML transforming ability” is not clear. Increased clonogenic potential in serial replating does not equal AML transformation.

2. F4I: Please show pSTAT5 and STAT5

3. In F5E-F, the shJak2/Nras wt control is missing

4. Please clarify and define the term “molecular follow-up” as it is frequently mentioned in the manuscript without definition.

Reviewer #5

(Remarks to the Author)

Version 1:

Reviewer comments:

Reviewer #1

(Remarks to the Author)

I have no further comments

Reviewer #2

(Remarks to the Author)

The authors have addressed all important issues of the original review.

Reviewer #3

(Remarks to the Author)

The authors have sufficiently addressed my minor concerns.

Reviewer #4

(Remarks to the Author)

The authors have taken a comprehensive approach to address my concerns. The manuscript has been improved very

substantially. I have no further comments.

Reviewer #5

(Remarks to the Author)

RESPONSE TO THE REVIEWERS

We would like to thank the Reviewers for their overall support of the manuscript and constructive suggestions.

We have added to the former figures and supplements: a comprehensive re-analysis of our clinical and molecular cohort data, and 14 new experiments including experiments with primary patient samples and newly generated *in vitro* models. These new experiments were designed to I) strengthen the statistical association between ruxolitinib exposure and selection of *RAS* mutated clones and its impact on patients prognosis, II) evaluate the effect of ruxolitinib exposure on the selection of diverse types of *RAS* mutations and on *ASXL1* mutations and III) highlight the impact of ruxolitinib-mediated *RAS* clonal selection in a non *JAK2* driver context. These results are now presented in the new **Figure sub-panels 1H, 2F, 3A, 4L-M, 6G, S1B-E, S2, S3C-F, S4A-F, S5A-B** and **Tables S2, S3, S5, S6, S9, S10**.

In aggregate, we believe that *Nature Communications* is an ideal forum to report these new findings highlighting the selection of *RAS* mutated clones upon exposure to the *JAK2* inhibitor ruxolitinib and its clinical implications. Our results stem from a comprehensive clinical and translational study establishing *in vitro* and *in vivo* the causal link between ruxolitinib and the selection of *RAS* clones. They have direct clinical implications for the management of patients with Myeloproliferative Neoplasms.

Our responses to the specific comments are detailed as below:

Reviewer #1:

In their manuscript entitled “*JAK2* inhibition mediates clonal selection of *RAS* pathway mutations in myeloproliferative neoplasms”, Maslah et al. explore the impact of *RAS* mutations in the setting of ruxolitinib treatment for MPN. Using various levels of evidence (patient data, patient derived single cell colonies, murine experiments, cell culture experiments), they uncover an unexpected mechanism of selection of *RAS* mutated clones induced by *JAK2* inhibition. The finding itself is unexpected, interesting and worth reporting. I appreciate the elaborate, multi-dimensional approach including clinical and pre-clinical data. However, in its current version, the manuscript has various weaknesses. In particular, I feel that some conclusions are premature and warrant broader investigation.

We thank the reviewer for his overall enthusiasm and for highlighting the importance of the comprehensive clinical to pre-clinical approach developed in our study.

Major points:

1) The introduction is well written but information can be shortened to a maximum of approx.. 1.5 pages

The introduction has been shortened according to the reviewer’s suggestion.

2) Most problematic about comparing ruxo-treated vs. untreated patients is that they are clinically different cohorts with different risk profiles. As shown in table S1, diseases in the non-treated cohort are in earlier clinical stages. Comparisons between these groups should therefore be avoided, in particular with regard to leukemic transformation and OS. Therefore, I would suggest to perform analyses as follows: stratification according to DIPSS at molecular evaluation: group 1:

low/intermediate-1 group 2: intermediate-2/high. Further analysis according to RAS mutational status and ruxo yes/no.

We agree with the reviewer that comparing ruxo-treated and untreated patients encompasses a number of confounding biases related to the clinical use of ruxolitinib in patients with more aggressive disease. This is the main reason why we validated the ruxolitinib-mediated RAS clonal selection using a variety of *in vitro* and *in vivo* models.

Nonetheless, we agree that providing additional statistical analysis of our patients' cohort outcomes is key for the readers' interpretation of our findings. Therefore, data has been re-analyzed after stratification according to DIPSS at molecular evaluation prior to evaluating the impact of RAS mutational status on OS and TFS among patients treated or not with ruxolitinib. Unfortunately, a low number of RAS mutations and death/transformation events in the DIPSS low/intermediate-1 group precluded statistical analysis. The new data is now presented in **Figure S1E** and discussed in the manuscript results section as follows:

“Given the intrinsic biases associated with comparing ruxolitinib treated and non-treated patients, due to the clinical use of ruxolitinib in patients with more aggressive disease, we also evaluated the impact of RAS mutational status on OS and TFS after stratification according to the DIPSS score. Importantly, presence of RAS mutations remained associated with a poorer OS and TFS among patients harboring an intermediate-2/high risk score when exposed to ruxolitinib (HR 3.8 CI95%[1.3; 11.0], p=0.012 and HR 6.2 CI95%[1.6; 24.2], p=0.009) (Figure S1E). Conversely, these associations were not significant among intermediate-2/high risk score patients not exposed to ruxolitinib (HR 1.6 CI95%[0.2; 14.3], p=0.682 for OS and HR 2.0 CI95%[0.2; 19.5], p=0.559 for TFS). A low number of RAS mutations and death/transformation events precluded a similar analysis among DIPSS low/intermediate-1 patients.”

The discussion section has also been updated to inform the reader that our results are particularly true in the DIPSS int-2/high group: *“In terms of clinical implications for the management of patients with MPN, our results suggest that ruxolitinib-induced selection of RAS-mutated clones might negatively impact prognosis in patients with myelofibrosis, particularly in the DIPSS intermediate-2/high risk group.”*

“Given the low number of patients with RAS pathway mutations in the non-ruxolitinib treated patient group and in the DIPSS low/intermediate-1 risk categories in our study, our clinical findings need to be confirmed in larger external cohorts and across all DIPSS categories.”

In addition, according to Reviewer#4's suggestion, OS and TFS have also been evaluated in the full patient cohort prior to analyzing RAS mutations impact according to ruxolitinib exposure (**Figure S1D and Tables S5-6**).

3) On a related note, as only 6 patients in the untreated cohort have a RAS mutation, a more careful wording must be chosen when describing potential impact on TFS and OS. This is particularly true for the conclusion in the discussion (p.19). Increased leukemic transformation was for example not described in the long-term data of the COMFORT II trial. Please comment.

We thank the reviewer for his thoughtful comment and agree that our clinical findings regarding patient outcomes need to be further validated in larger external cohorts. We have thus softened our language and updated the discussion to include the fact that increased transformation was not observed in long-term analysis of phase 3 ruxolitinib trials. We hope that this will make our results interpretation clearer for the reader.

“However, we showed that poorer OS and TFS related to these mutations are mainly observed in patients treated with ruxolitinib, while OS and TFS did not seem to be statistically impacted in patients that did not

receive JAK2 inhibitor therapy. This observation suggests that the selective pressure induced by ruxolitinib could adversely impact patient outcome. Importantly, long-term findings from ruxolitinib phase 3 clinical trials did not report a higher incidence of secondary AML in the ruxolitinib treated group^{7,8,53,54}. However, such long-term clinical trials re-analysis harbors a number of biases driven by the fact that these studies were not designed for long-term randomized groups' comparison. Cross-over to the ruxolitinib arm, absence of longitudinal NGS molecular follow-up and the low rate of RAS mutations and leukemic transformation events in MPN, could have masked such observation. Given the low number of patients with RAS pathway mutations in the non-ruxolitinib treated patient group and in the DIPSS low/intermediate-1 risk categories in our study, our clinical findings need to be confirmed in larger external cohorts and across all DIPSS categories."

4) What was the mean (max.-min.) treatment duration with ruxolitinib until transformation?

Mean ruxolitinib treatment duration until transformation was 44.7 months _{Range}[4.2; 81.5] in the global cohort, 33.5 months _{Range}[4.2; 81.5] for patients harboring a RAS mutation and 62.5 months _{Range}[36.7; 77.3] for patients without RAS mutation. This information has been added to the manuscript results section.

5) Did patients undergo allogeneic stem cell transplantation and if so, how was this data dealt with in the OS/TFS analysis?

Within our cohort, 8 patients had allogeneic stem cell transplantation (5/71 patients in the non ruxolitinib group and 3/72 patients in the ruxolitinib group). Our OS/TFS data have now been re-analyzed censoring these patients at time of bone marrow transplantation. This did not significantly impact our results; however, the manuscript was updated with the new results (**Figures 2B-C and S1 D-E and Tables S5-S8**) and this was specified in the methods section.

6) A general problem of the manuscript is that all types of NRAS/KRAS/HRAS mutations are set equal. This confers to the patient data as well as to the preclinical part.

- **The manuscript provides no information on the individual mutations. This could be addressed by a lollipop plot, for example. Were the mutations only hotspot mutations or were there also non-hotspot mutations found?**

As suggested by the reviewer, lollipop plots have now been added to detail RAS mutations (**Figure 1H**). RAS mutations type and VAF are now also detailed in **Table S3**, reporting NGS data among the global study cohort as suggested by this reviewer in point 12. NRAS/KRAS mutations were mainly hotspot mutations.

- **Please explain the choice of RAS mutations analyzed in the competition experiments (Figure 4). Why was NrasG12D tested in the first part of the co-cultures whereas NrasQ61K vectors were used in the second part? Was this due to a scientific rationale or due to the availability of the models?**

The choice of RAS mutations was due to 1) model availability and 2) validation of ruxolitinib mediated clonal selection in 2 different *Nras* mutational backgrounds and *in vivo* models. *Nras*^{Q61K} plasmids were available for retroviral transduction and *Nras*^{G12D} cells were available from hMRP8-*Nras*^{G12D} mice. However, according to this reviewers' comment, we have now included other RAS mutations in our *in vitro* studies (see below).

Though it is not feasible to analyze all types of NRAS/Nras/KRAS/Kras mutations in the murine setting, the authors should at least discuss this limitation of their study. Though the biological impact of

different RAS-mutations seems to point into the same direction, differences between the different hotspot mutations and isoforms are described (e.g. different effects on downstream signaling/interactions). However, in the cell culture setting (experiments depicted in Figure 6), testing at least a second NRAS hotspot mutation or KRAS mutation would greatly increase the meaningfulness and reproducibility of the study.

We generated *NRAS*^{G12V}, *NRAS*^{G12D} and *KRAS*^{G13D} HEL expressing cells to evaluate the effect of ruxolitinib on clonal selection of different RAS isoforms and mutations. While RAS WT cell growth was impaired after 6 days of ruxolitinib treatment, *NRAS*^{G12V}, *NRAS*^{G12D} and *KRAS*^{G13D} HEL cells were resistant to JAK/STAT inhibition (Figures S3C-D). In our *in vitro* competition assay, ruxolitinib treatment also resulted in the positive selection of *NRAS*^{G12V} and *KRAS*^{G13D} HEL cells (Figure S3E).

Moreover, we have now evaluated *ex vivo* clonal selection under ruxolitinib exposure in 2 additional primary patient samples (n=6 total). Altogether, our primary patient data shows selection of 3 different *NRAS* mutations and 1 *KRAS* mutation (Figure 3A).

These new data therefore validate the impact of ruxolitinib on clonal selection of diverse RAS mutant cells. Finally, as suggested by the reviewer, the potential impact of different RAS isoforms and mutations is now discussed in the manuscript. *“Importantly, although our study highlights the impact of ruxolitinib on clonal selection of diverse RAS mutant cells, isoform- and mutation-specific differences in protein structure and signaling are described and further studies are required to precise potential differences upon ruxolitinib exposure.”*

7) All experiments were performed in the setting of JAK2 V617F, the interaction between CALR and MPL was not explored. This limitation should be discussed.

We agree with the reviewer that this is an important point to address, particularly since our clinical and primary patient data points to the selection of RAS mutated clones independently from the driver context. Indeed, Figure 3A now includes data on CD34⁺ cells extracted from 6 MPN patients harboring *JAK2*^{V617F} (n=4) but also *CALR* type 1 (n=1) and *MPL*^{W515L} (n=1) mutations. Single cell DNA-sequencing data for patients 2 and 4 shows that the RAS mutant selected cells are present within the *CALR* or *MPL* driver clone (Figure 3B). Additionally, we used Ba/F3 cells expressing exogenous *MPL* and *CALR*-del52 (Ba/F3-*MPL*-*CALR*del52 cell line) to evaluate the effect of ruxolitinib on RAS mutations clonal selection within a *CALR* mutated context (Figure 4L). While RAS WT cell growth was impaired after ruxolitinib treatment, Ba/F3-*MPL*-*CALR*del52 *NRAS*Q61K cells were resistant to JAK/STAT inhibition (Figure S3F). In our *in vitro* competition assay, ruxolitinib treatment resulted in the positive selection of Ba/F3-*MPL*-*CALR*del52 *NRAS*Q61K cells (Figures 4M).

These new data therefore experimentally validate the impact of ruxolitinib on clonal selection of RAS mutant cells in a *CALR* driver context *in vitro*.

Additionally, as suggested by the reviewer, we now also discuss this limitation in the manuscript: *“Importantly, while our primary patient and in vitro data points towards a similar RAS clonal selection across patients harboring the three MPN driver mutations, the majority of our validation experiments were conducted within a JAK2^{V617F} mutational context and further experimental data is required to more precisely evaluate the interaction between RAS and CALR/MPL mutations.”*

8) Role of ASXL1: Figure 1B shows a relevant increase in the number of ASXL1 mutations in ruxo-treated patients and Figure 2F shows that 7/8 transformed cases had an ASXL1 mutation. The role of this mutation is not investigated or discussed in the manuscript, though when looking at the pie charts, it

also appears to be positively selected. Did VAF increase during transformation?

We thank the reviewer for his suggestion and agree that studying the impact of ruxolitinib on clonal selection of *ASXL1* is important to evaluate the specificity of the reported RAS clonal selection. We have now evaluated *ASXL1* mutations variant allele frequency between baseline and follow-up NGS according to ruxolitinib exposure (**Figures S1B-C**), as well as at time of transformation (**Figure 2F**).

These new data suggests a specific effect of ruxolitinib on RAS pathway mutations selection and is now presented in the manuscript results:

“Importantly, while 10 *ASXL1* mutations (n=10 /53, 19%) were newly detected among ruxolitinib exposed patients in comparison with only 1 acquired *ASXL1* mutation (n=1/ 14, 7%) in the non-ruxolitinib exposed population (**Figures 1B-C**); *ASXL1* mutations variant allele frequency did not increase at follow-up molecular evaluation (**Figures S1B-C**). This observation suggests a specific effect of ruxolitinib on RAS pathway mutations clonal selection.”

*“While 7 of these 8 patients harbored an *ASXL1* mutation at time of transformation, including the patient who transformed in the absence of ruxolitinib exposure, variant allelic frequency of *ASXL1* mutations only increased in 1 patient (**Figure 2F**). This finding argues again for the specific effect of ruxolitinib on RAS pathway mutations selection.”*

Moreover, according to Reviewer#4 suggestion, we also performed additional *in vitro* experiments to evaluate the effect of ruxolitinib exposure on *ASXL1* mutation selection. Given the controversy around the gain or loss-of function impact of *ASXL1* mutations in MPN, we developed two complementary strategies to evaluate the impact of ruxolitinib on clonal selection of HEL cells 1) overexpressing an *ASXL1*^{G646Wfs*12} mutation, or 2) expressing shRNAs targeting *ASXL1*. In both cases, ruxolitinib exposure only mildly remodeled the fitness of the *ASXL1* clone (**Figures S4A-F**).

Minor points:

9) p.2: I think it is not shown that the RAS mutations are acquired by ruxo therapy. Probably, the term “clonal outgrowth” is more accurate.

We agree with the reviewer that ruxolitinib is associated with clonal selection/outgrowth rather than acquisition of RAS mutations. This has been updated in the revised abstract.

10) Table S1: Information on missing data should be included (i.e. data lacking for IPSS, DIPSS, etc.). Information on missing data is now included in the revised **Table S1**.

11) Did the NGS panel only cover RAS mutational hotspots or the whole gene? Please provide the respective information, preferably for all genes investigated.

A list of genes and exons covered in our NGS panel is now provided in **Table S10**. Only exons 2 and 3 of *KRAS* and *NRAS* which include RAS mutational hotspots were covered.

12) Include results of NGS analysis (mutation, VAF, etc.) for each patient (table in supplement) Mutations identified using our NGS panel and their respective VAFs are now detailed in **Table S3**, according to ruxolitinib exposure.

Reviewer #2 (Remarks to the Author):

This is a very interesting study, in which the authors examined the effect of ruxolitinib on the clonal evolution of MPN, specifically myelofibrosis. Single-cell DNA sequencing combined with ex vivo treatment of RAS mutated CD34+ primary cells, demonstrated that ruxolitinib induced RAS clonal selection. Such clonal selection was associated with decreased transformation-free and overall survival in patients treated with ruxolitinib. They also found that MAPK pathway activation was associated with JAK2 downregulation which resulted in enhanced oncogenic potential of RAS mutations. This work has important implications on our overall understanding of the mechanisms of leukemogenesis and may prove to have significant impact on the clinical management of MPNs. In general, the studies have been carefully done and the results are detailed, clear and convincing.

We thank the reviewer for the strong support of our manuscript.

Some minor suggestions are summarized below:

1. Perhaps the authors should include a statement about cell line authentication for the MPN derived cell lines.

The following statement has been added to the methods: *“All post-MPN AML cell lines were confirmed by STR genotyping.”*

2. In page 18, first sentence, the authors most likely meant to write “ineffective” instead of “effective”. We meant that in specific mutational contexts, activation of oncogenic pathways can paradoxically decrease cellular fitness. To avoid any ambiguous interpretation, the initial sentence was deleted to keep only the following: *“Our results complement these findings and highlight in patients the counter-intuitive oncogenic risk of impairing oncogenic pathways in specific mutational contexts.”*

3. Deposition and accessibility of the sequencing data from patients. This will obviously be important for future investigations. It was not clear in the text how these will be made available and this should be included.

We are committed to make our data available for the community. As also suggested by Reviewer#1, mutations identified using our NGS panel and their respective VAFs are now detailed in **Table S3**, according to ruxolitinib exposure.

Reviewer #3 (Remarks to the Author):

This is an important, comprehensive and well-written study that reveals a potential mechanism underlying the development of ruxolitinib resistance in MPN patients. Maslah et al. demonstrate that ruxolitinib treatment induces selection of RAS mutant clones in both wild type cells and MPN cells with constitutive JAK/STAT activation. Interestingly, the authors show that the mechanism underlying this phenomenon relates to MAPK-mediated downregulation of JAK2 expression such that JAK/STAT activation is “fine-tuned” to support MAPK activation dosage. Overall, the findings are well supported, and this study represents an important advancement in the MPN field.

We thank the reviewer for the strong support of our manuscript.

I have only minor suggestions that I believe will strengthen an already strong study:

1. The findings in Figure 3 are interesting, but would be more robust with the addition of more MPN patients (8-10 would be ideal).

We have now included primary patient cell culture data for 2 additional patients (total of 6 patients) presented in **Figure 3A** and **Figure S2**. Unfortunately, as RAS mutations remain rare and these experiments require fresh CD34⁺ sorted cells, we were not able to obtain more samples within the revision timeframe. Nonetheless, the results from the 2 additional patients are consistent with our previous results, confirming clonal selection of RAS mutated cells under ruxolitinib exposure in a total number of 6 primary patient samples harboring diverse genetic backgrounds.

2. The cell line and mouse assays focus on Jak2 V617F – these findings would be greatly strengthened by the addition of models for mutant CALR and MPL, to show this is generally true across driver oncogenes that hyperactivate JAK/STAT signaling.

We agree with the reviewer that this is an important point to address, particularly since our clinical and primary patient data, points to the selection of RAS mutated clones independently from the driver context. Indeed, **Figure 3A** now includes data on CD34⁺ cells extracted from 6 MPN patients harboring *JAK2*^{V617F} (n=4) but also *CALR* type 1 (n=1) and *MPL*^{W515L} (n=1) mutations. Single cell DNA-sequencing data for patients 2 and 4 shows that the RAS mutant selected cells are present within the *CALR* or *MPL* driver clone (**Figure 3B**). Additionally, we used Ba/F3 cells expressing exogenous MPL and CALR-del52 (Ba/F3-MPL-CALRdel52 cell line) to evaluate the effect of ruxolitinib on RAS mutations clonal selection within a CALR mutated context (**Figure 4L**). While RAS WT cell growth was impaired after ruxolitinib treatment, Ba/F3-MPL-CALRdel52 NRASQ61K cells were resistant to JAK/STAT inhibition (**Figure S3F**). In our *in vitro* competition assay, ruxolitinib treatment resulted in the positive selection of Ba/F3-MPL-CALRdel52 NRASQ61K cells (**Figures 4M**).

These new data therefore experimentally validate the impact of ruxolitinib on clonal selection of RAS mutant cells in a CALR driver context *in vitro*.

Additionally, as suggested by Reviewer#1, we now also discuss this limitation in the manuscript: *“Importantly, while our primary patient and in vitro data points towards a similar RAS clonal selection across patients harboring the three MPN driver mutations, the majority of our validation experiments were conducted within a *JAK2*^{V617F} mutational context and further experimental data is required to more precisely evaluate the interaction between RAS and CALR/MPL mutations.”*

Reviewers #4 & #5 (Remarks to the Author):

Maslah and colleagues have investigated the role of RAS pathway mutations in myelofibrosis patients treated with ruxolitinib (rux) or not. Their key finding is that rux favor the outgrowth of RAS-mutated clones, and that rux impairs the outcome of patients with RAS pathway mutations. A series of in vitro and in vivo competition experiments show that reduced JAK/STAT signaling, either by rux or by JAK2 knockdown, promotes mutant RAS-mediated cellular fitness and oncogenesis, apparently by partially blocking the senescence induced by mutant RAS. This paper has a number of interesting aspects, but also several profound weaknesses, particularly with respect to novelty and the COX model.

We thank the reviewer for his thorough review of our manuscript, pointing the interesting aspects of our findings. We hope that our detailed answer below addresses the reviewers' points and believe that integration of this suggestions strengthens the reported findings.

Major concerns

1. Novelty is limited. RAS pathway activating mutations have been described as resistance drivers in MPN and in FLT3-ITD-positive AML, IDH1/2-mutated AML. The mutual exclusivity of STAT and RAS/MAPK signaling has been reported for ALL by the Mueschen lab. All these are acknowledged by the authors, but the fact remains. One would like to see mechanistic studies that go beyond the descriptive level. For example, is the pro-oncogenic effect of inhibiting JAK/STAT mediated by reducing the specific MAPK output emanating from JAK/STAT or by blocking a set of pSTAT3 or pSTAT5 target genes?

The substantial novelty of our study lies in two key aspects. First, we provide conclusive evidence of a statistical association and comprehensively demonstrate for the first time a causal link between exposure to ruxolitinib and the clonal selection of RAS mutations. Importantly, we show that this effect is on target and stems from JAK2 inhibition, thereby uncovering a clinical concern regarding the utilization of other FDA-approved JAK2 inhibitors in the treatment of MPN patients. Second, our study reveals, for the first time that the prognostic impact of RAS mutations in MPN is mainly observed in the context of ruxolitinib exposure. We elucidated mechanistically this phenomenon by unveiling a ruxolitinib-driven senescence release phenotype. This not only identifies a previously unrecognized aspect of RAS mutations but also opens up avenues for innovative therapeutic alternatives tailored for RAS-mutated MPN patients. Specifically, we now propose a rationally designed option: the combination of ruxolitinib with trametinib. We believe that these findings mark a significant advancement in understanding and treating RAS-mutated MPN.

We agree with the reviewer that our findings open the way for further mechanistic studies. To evaluate whether the pro-oncogenic effect of inhibiting JAK/STAT is mediated by reducing the specific MAPK output emanating from JAK/STAT or by blocking a set of pSTAT3 or pSTAT5 target genes, we used two small molecule chemical inhibitors of STAT3 and STAT5 SH2 domains (Stattic and AC-4-130), either alone or in combination. STAT3 inhibition resulted in a slightly lower decrease in cellular fitness of *NRAS*^{Q61K} cells, in comparison with non-RAS mutated cells. STAT5 inhibition resulted in a mild increase in cellular fitness of *NRAS*^{Q61K} cells, in comparison with non-RAS mutated cells. Dual STAT3/5 inhibition within *NRAS*^{Q61K} cells partially rescued its anti-proliferative effect observed on non-RAS mutated cells. These results suggest that ruxolitinib RAS clonal selection might at least partially result from 1) inhibition of STAT3 downstream targets, and/or 2) inhibition of downstream redundant targets of STAT3/5 that would require inhibition of both STATs to observe an effect.

Growth inhibition of HEL human AML cell lines expressing either an Empty or a NRAS^{Q61K} encoding vector after two to four days of STAT3 inhibitor (Stattic, 180 nM), STAT5 inhibitor (AC- 4-130, 1.5 μM) or a combination of both treatments. Statistical significance determined using Welch's t-test in comparison to DMSO (*) or to Empty vector condition (#). Error bars represent mean of 8 to 10 replicates ± SD

However, we have not included this new data in the manuscript as we believe that this approach needs to be complemented with genetic gain and loss of function approaches to obtain a definitive answer on the role of MAPK vs STAT3/5 pathways on ruxolitinib induced RAS clonal selection.

We believe that this is beyond the scope of the current manuscript which already includes an epidemiological longitudinal study on a large cohort of MPN patients, a large set of in vitro experiments including experiments on primary patient samples and in vivo models validating the causal link between ruxolitinib exposure and RAS pathway clonal selection.

2. There are concerns about the COX model. It does not make sense that HMR mutations are significantly associated with OS but not TFS, suggesting that the model is unstable.

The Hazard Ratio for HMR mutations association with TFS could not be calculated due to the fact that none of the 26 ruxolitinib treated patients without HMR transformed into AML/MDS. This does not mean that there is no association but represents a statistical limitation intrinsically related to our cohort data. Indeed, 13 patients among the 46 ruxolitinib treated patients harboring HMR mutations transformed to AML/MDS. We thank the reviewer for pointing this and have updated **Table S8** and the corresponding legend to clarify this for the reader.

Additionally, in the global cohort analysis performed upon this reviewers' suggestion and detailed below, HMR mutations were significantly associated with TFS (**Table S6**).

What were the causes of death in the non-rux group?

A total of 9 patients died within the non-ruxolitinib treated group. Causes of death are now detailed in **Table S9**, they include hemorrhage, infection and global condition deterioration, with 4 patients who died in an AML/MDS disease stage while 5 died in chronic MPN.

How were these patients treated?

In the non-ruxolitinib arm, 45% (32/71) patients did not receive any cytoreductive treatment. The majority of the remaining patients received hydroxyurea, pegylated interferon alfa-2a or a combination of both. Treatments received by this population are now detailed in **Table S2**.

The individual components of the IPSS should be included as well as cytogenetics.

Table S1 has been updated according to reviewer's suggestion to include individual components of IPSS and cytogenetic data, according to ruxolitinib treatment status. Ruxolitinib treated patients had a slightly lower hemoglobin level (11.25 vs 12 g/dl), and more constitutional symptoms (32% vs 15%), in agreement with ruxolitinib indications. This information is now also described in the results section of the manuscript.

Was the year of diagnosis different between the non-rux and rux cohorts?

Non-ruxolitinib treated patients were diagnosed between 1990 and 2019 while ruxolitinib treated patients were diagnosed between 1994 and 2018. 73% (n=52/71) and 47% (n=34/72) patients were diagnosed after ruxolitinib EMA approval for myelofibrosis (in 2012 or later), in the non-ruxolitinib and ruxolitinib treated groups respectively. This information has now been added to the manuscript.

It is unclear why OS and TFS as not evaluated for the entire cohort in the first place, with RAS mutations and ruxolitinib included as variables, then analyzed for significant interactions. Could propensity matching be used to reduce the imbalances between the rux and non-rux groups?

We agree with the reviewer that comparing ruxo-treated and untreated patients encompasses a number of confounding biases related to the clinical use of ruxolitinib in patients with more aggressive disease. This is the main reason why we validated the ruxolitinib-mediated RAS clonal selection using a variety of *in vitro* and *in vivo* models.

Nonetheless, we agree that providing additional statistical analysis of our patients' cohort outcomes is key for the readers' interpretation of our findings. Therefore, according to the reviewers' suggestion, we re-analyzed our data to evaluate the impact of RAS mutations on OS/TFS in the global cohort prior to analyzing RAS mutations impact according to ruxolitinib exposure.

"In the overall cohort, presence of RAS mutations was associated with decreased OS and TFS in univariate analysis (HR 3.2 CI_{95%}[1.6; 6.4], p=0.001 and 5.2 CI_{95%}[2.1; 12.5], p<0.001). RAS mutational status remained significantly associated with TFS but not with OS in the multivariate analysis (3.0 CI_{95%}[1.2; 7.8], p=0.023 and p=0.0884) (Figure S1D and Tables S5-6)."

Additionally, although propensity matching would have been a good option to reduce imbalances between ruxolitinib and non-ruxolitinib treated groups, this analysis was not feasible within our cohort as it would require a larger number of patients to allow for matching on different variables. Alternatively, upon suggestion of Reviewer#1, data has been re-analyzed after stratification according to DIPSS at molecular evaluation prior to evaluating the impact of RAS mutational status on OS and TFS among patients treated or not with ruxolitinib. The new data is now presented in **Figure S1E** and discussed in the manuscript results section as follows:

"Given the intrinsic biases associated with comparing ruxolitinib treated and non-treated patients, due to the clinical use of ruxolitinib in patients with more aggressive disease, we also evaluated the impact of RAS mutational status on OS and TFS after stratification according to the DIPSS score. Importantly, presence of RAS mutations remained associated with a poorer OS and TFS among patients harboring an intermediate-2/high risk score when exposed to ruxolitinib (HR 3.8 CI_{95%}[1.3; 11.0], p=0.012 and HR 6.2 CI_{95%}[1.6; 24.2], p=0.009) (Figure S1E). Conversely, these associations were not significant among intermediate-2/high risk score patients not exposed to ruxolitinib (HR 1.6 CI_{95%}[0.2; 14.3], p=0.682 for OS and HR 2.0 CI_{95%}[0.2; 19.5], p=0.559 for TFS). A low number of RAS mutations and death/transformation events precluded a similar analysis among DIPSS low/intermediate-1 patients."

The discussion section has also been updated to inform the reader that our results are particularly true in

the DIPSS int-2/high group: *“In terms of clinical implications for the management of patients with MPN, our results suggest that ruxolitinib-induced selection of RAS-mutated clones might negatively impact prognosis in patients with myelofibrosis, particularly in the DIPSS intermediate-2/high risk group.”*

“Given the low number of patients with RAS pathway mutations in the non-ruxolitinib treated patient group and in the DIPSS low/intermediate-1 risk categories in our study, our clinical findings need to be confirmed in larger external cohorts and across all DIPSS categories.”

Validation of the results in an independent cohort is critical.

We agree with the reviewer that validation of our clinical results in an external cohort is important. However, since longitudinal molecular follow-up is not routinely performed in MPN, we were not able to access large clinically and molecularly annotated myelofibrosis cohorts with longitudinal molecular data available within the timeframe of this manuscript revision. We hope that publication of our manuscript will encourage generation of such longitudinal data in large cohorts to externally validate our results, ideally in a prospective manner. Nevertheless, our results were validated using a large set of preclinical *in vitro* and *in vivo* models to limit the biases associated with epidemiological studies.

This limitation is now discussed in the manuscript: *“Given the low number of patients with RAS pathway mutations in the non-ruxolitinib treated patient group and in the DIPSS low/intermediate-1 risk categories in our study, our clinical findings need to be confirmed in larger external cohorts and across all DIPSS categories.”*

3. More details on treatment and response should be provided: (i) Please comment on specific aspects of ruxolitinib therapy such as treatment dose, duration of exposure, treatment interruptions and whether they were associated with clinical endpoints (not only survival, but also others as mentioned below).

Detailed information on treatment dose and its association with clinical endpoints was not available for our cohort. Although this variable is complicated to assess in a retrospective manner, all our patients were followed in a specialized reference center and ruxolitinib dose adjustments were adapted to their tolerance mainly in terms of cytopenias according to the international recommendations. This is now discussed as a limitation: *“Detailed information on ruxolitinib dose was not available for our retrospective cohort, precluding analysis of its potential impact on clinical outcomes. This should be further evaluated in future studies to inform on management of patients harboring RAS pathway mutations.”*

Median ruxolitinib exposure was 16 months $_{IQR}[12; 28]$ prior to last molecular evaluation in which RAS pathway mutations were identified (**Figure S1A**). Mean ruxolitinib treatment duration until transformation was 44.7 months $_{Range}[4.2; 81.5]$ in the global cohort, 33.5 months $_{Range}[4.2; 81.5]$ for patients harboring a RAS mutation and 62.5 months $_{Range}[36.7; 77.3]$ for patients without RAS mutation. This is now detailed in the manuscript.

(ii) Comment on whether patients initially had a clinical response to ruxolitinib and subsequently lost response upon development of RAS pathway alteration vs. they never responded to treatment even before development of RAS pathway alteration.

We thank the reviewer for this suggestion and agree that this is informative for the reader. Among the 8 patients with newly acquired RAS pathway mutations within the ruxolitinib treated cohort, 6 initially responded to ruxolitinib treatment (5 clinical improvement (CI) and 1 partial response (PR)), while 2 did not respond (1 stable disease (SD) and 1 progressive disease (PD)). This is now discussed in the manuscript.

(iii) Is there information available on subsequent therapies patients received upon development of RAS

alterations (continued vs. stopped ruxolitinib, switched or added other therapies) as they could influence overall survival and transformation free survival.

Among the 8 patients with newly acquired RAS pathway mutations within the ruxolitinib treated cohort, 5 continued on ruxolitinib treatment after identification of the RAS mutation while ruxolitinib was interrupted for the 3 others (no cytoreductive treatment for 1 patient, hydroxyurea and RBC transfusion for 1 patient and splenectomy for the third one). Given the retrospective nature of our study, patients' treatment was not specifically adjusted after identification of RAS pathway mutations as their prognosis impact was not known at the time and very limited alternative therapies were available for these patients. Treatment was only adapted based on patients' response to ruxolitinib. This is now discussed in the manuscript.

4. NGS: Molecular follow-up is available only for half of the cohort.

Although molecular follow-up is not routinely recommended for all myelofibrosis patients, it was implemented within our expert center to explore clonal evolution dynamics and its impact on patients' outcome. This implementation is more recent than performing NGS at diagnosis which is now routinely done for all myelofibrosis patients in many center. Therefore, longitudinal data is only available for part of the cohort. We believe that our study will open the way to a more generalized use of molecular follow-up in myelofibrosis patients' management.

I could not find the list of genes included in the panel. Does it include PTPN11? BRAF? What is the sensitivity of the assay?

Our NGS panel covered 36 genes involved in myeloid malignancies listed in the methods section. These include *PTPN11* and *BRAF* but no mutations were found in these genes across our cohort. Interestingly, ruxolitinib *ex vivo* treatment of our primary patient samples revealed a *PTPN11*^{Q510E} low VAF mutation that was not detected prior to ruxolitinib (Patient#2, **Figure S2**). Although this might point to a potential selection of *PTPN11* mutations, further studies in larger cohorts are required to explore the role of ruxolitinib on *PTPN11* given the low incidence of such mutations in MPN. A sentence has been added in the results section to point this observation: *"Of note, although no mutation was found in the RAS pathway regulator PTPN11 across our cohort, ruxolitinib ex vivo treatment of our primary patient samples revealed a PTPN11^{Q510E} low VAF mutation that was not detected prior to ruxolitinib (Patient#2, Figure S2)."*

The list of genes and covered regions included in our NGS panel is now also reported in **Table S10**.

The sensitivity of our NGS is 1%, this is stated in the methods section of the revised manuscript.

Since the data are very limited for some of the analysis, it is unclear what we can conclude from it – e.g. the molecular landscape of AML (N=1 for non-rux; n = 7 for rux).

We agree with the reviewer that data at time of transformation is limited given the low number of events, this data is therefore only provided to inform the reader in a descriptive manner. The corresponding panel has now been moved to **supplementary Figure S1F** instead of main **Figure 2F**.

We thank the reviewer for pinpointing this important details on our molecular analysis and believe that the added information will make interpretation of our results clearer for the reader.

5. In the experiment shown in F4C, it is unclear why Jak2WT/NrasG12D was competed against Jak2V617F/NrasWT and not Jak2V617F/NrasWT vs Jak2V617F/NrasG12D. The system represents one equation with 2 variables and cannot be solved. This is essentially what was done in F4E/F.

The *in vivo* experiments in **Figures 4C-F** were designed to mimic the two clonal selection situations

identified using single cell DNA sequencing in **Figure 3C**. Indeed, the first *in vivo* experiment (**Figures 4C-D**) represents patients in which the *RAS* mutation was present in a clone without a driver mutation. In this setting, Jak2^{WT}/Nras^{G12D} cells are in competition with Jak2^{V617F}/Nras^{WT}. The second experiment (**Figures 4E-F**) represents patients in which the *RAS* mutation was present within the MPN driver clone. In this setting, Jak2^{V617F}/Nras^{Q61K} cells are in competition with Jak2^{V617F}/Nras^{WT} cells.

We thank the reviewer for highlighting this and have now updated the results section to make our rationale for this experimental design clearer for the reader.

6. It would strengthen the manuscript if the authors included a control consisting of an unrelated oncogene, e.g. mutant ASXL1, and showed that ruxolitinib does not select for cells expressing this variant.

We thank the reviewer for his suggestion and agree that studying the impact of ruxolitinib on clonal selection of *ASXL1* is important to evaluate the specificity of the reported *RAS* clonal selection. We have now performed additional *in vitro* experiments to evaluate the effect of ruxolitinib exposure on *ASXL1* mutations selection. Given the controversy around the gain or loss-of function impact of *ASXL1* mutations in MPN, we developed two complementary strategies to evaluate the impact of ruxolitinib on clonal selection of HEL cells 1)overexpressing an *ASXL1*^{G646Wfs*12} mutation, or 2)expressing shRNAs targeting *ASXL1*. In both cases, ruxolitinib exposure only mildly remodeled the fitness of the *ASXL1* clone (**Figures S4A-F**).

Additionally, as suggested by Reviewer#1, we have now also evaluated *ASXL1* mutations variant allele frequency between baseline and follow-up NGS according to ruxolitinib exposure (**Figures S1B-C**), as well as at time of transformation (**Figure 2F**). These new data suggests a specific effect of ruxolitinib on *RAS* pathway mutations selection and is now presented in the manuscript results:

“Importantly, while 10 *ASXL1* mutations (n=10 /53, 19%) were newly detected among ruxolitinib exposed patients in comparison with only 1 acquired *ASXL1* mutation (n=1/ 14, 7%) in the non-ruxolitinib exposed population (**Figures 1B-C**); *ASXL1* mutations variant allele frequency did not increase at follow-up molecular evaluation (**Figures S1B-C**). This observation suggests a specific effect of ruxolitinib on *RAS* pathway mutations clonal selection.”

“While 7 of these 8 patients harbored an *ASXL1* mutation at time of transformation, including the patient who transformed in the absence of ruxolitinib exposure, variant allelic frequency of *ASXL1* mutations only increased in 1 patient (**Figure 2F**). This finding argues again for the specific effect of ruxolitinib on *RAS* pathway mutations selection.”

7. Some of the observations are entirely expected. For instance, on p.11, F4I: “Interestingly, as opposed to the *RAS* WT cellular context, persistent ERK phosphorylation was observed upon ruxolitinib exposure in a *RAS* mutated context (Figure 4I).” Along the same lines, there is a lot of unnecessary redundancy, particularly in the discussion which should be shortened considerably.

This sentence has been removed in the results section as suggested by the reviewer.

Additionally, the discussion has been shortened avoiding redundancy while including the additional points to discuss suggested by all the reviewers.

8. I disagree with the interpretation of the cell cycle data (F6C-E). First, the subG1 cells must be removed – these are dead cells. Second, increased cell cycle entry should be manifest by more cells in S not in G2/M.

We initially included subG1 cells proportion as a surrogate of dead cells to show the differential effect of ruxolitinib on RASWT and RAS mutant cells. Upon reviewers' suggestion, we have now removed this cellular population to avoid any confusion.

We believe that S and G2/M cells can both be increased if cells re-engage into the cell cycle depending on which checkpoints are impaired/re-activated. To complement our findings and more directly evaluate cell cycle entry, we have now evaluated in an additional experiment *de novo* DNA synthesis during the S-Phase of the cell cycle using an EdU assay. These new results show decreased cell cycling upon ruxolitinib exposure in a RAS^{WT} context, while ruxolitinib resulted in an increased EdU uptake in an NRAS^{Q61K} context (**Figure 6G**). These results have been added to the manuscript to offer the reader an additional evaluation.

Minor concerns

1. The sentence on p.10 “Given the clinical impact of ruxolitinib-selected RAS mutations in our MPN patient cohort, we next sought to study its effects on the clonogenic potential of NrasQ61K cells, as a surrogate for their AML transforming ability” is not clear. Increased clonogenic potential in serial replating does not equal AML transformation.

We agree with the reviewer that increased clonogenic potential is not a definitive proof of transformation, but can rather be considered as part of a set of arguments arguing for potential transformation.

To avoid any confusion or over-statement, the sentence has been updated as follows: *“To further evaluate the impact of ruxolitinib on RAS clones oncogenic potential, we next sought to study its effects on the clonogenic potential of Nras^{Q61K} cells.”*

2. F4I: Please show pSTAT5 and STAT5

According to the reviewers' suggestion, an additional western blot has been performed complementing **Figure 4I** evaluating pSTAT5 and STAT5 levels. Overall pSTAT5 behaves similarly to pSTAT3.

3. In F5E-F, the shJak2/Nras wt control is missing

This control was not initially included in our experiment due to a limitation in the number of primary cells available, and as we expected from the literature that Jak2 knockdown in a RAS WT context would result in a decrease in the colony forming ability of hematopoietic cells which are known to be highly dependent on the JAK/STAT pathway.

According to the reviewers' suggestion and to offer the reader all the controls, we have now performed an additional colony formation experiment showing indeed that shJak2 results in decreased colony number both in a JAK2^{WT} and JAK2^{V617F} context (**Figure S5**).

4. Please clarify and define the term “molecular follow-up” as it is frequently mentioned in the manuscript without definition.

We thank the reviewer for catching this. Molecular follow-up is now defined in the methods as presence of at least 2 NGS molecular evaluations.

When this term was used improperly, we have now replaced it with “molecular evaluation” for more clarity.